



# Multiple modes of shoreline change along the Alaskan Beaufort Sea observed using ICESat-2 altimetry and satellite imagery

Marnie B. Bryant[1], Adrian A. Borsa[1], Claire C. Masteller[2], Roger J. Michaelides[2], Matthew R. Siegfried[3], Adam P. Young[1], and Eric J. Anderson[4]

[1]Institute of Geophysics and Planetary Physics, Scripps Institution of Oceanography, University of California San Diego, La Jolla, California 92093, USA
[2]Department of Earth, Environmental and Planetary Science, Washington University in St. Louis, St. Louis, Missouri 63130, USA
[3]Department of Geophysics, Colorado School of Mines, Golden, Colorado 80401, USA
[4]Hydrologic Science and Engineering, Colorado School of Mines, Golden, Colorado 80401, USA

**Correspondence:** Marnie Bryant (m1bryant@ucsd.edu)

**Abstract.** Arctic shorelines are retreating rapidly due to declining sea ice cover, increasing temperatures, and increasing storm activity. Shoreline morphology may influence local retreat rates, but quantifying this relationship requires repeat estimates of shoreline positions and morphologic properties. Here we use shoreline boundaries from multispectral imagery from Planet and topographic profiles from ICESat-2 satellite altimetry to compare year-to-year changes in shoreline position and morphology

across different shoreline types, focusing on an 8 km stretch of the Alaskan Beaufort Sea Coast during the 2019-2021 open water seasons. We consider temporal and spatial variability in shoreline change in the context of environmental forcings from ERA5 and morphologic classifications from the ShoreZone database. We find a mean spatially averaged shoreline change rate of -16.7 m/a over 3 years, with local estimates ranging from -70.1 m to +18.5 m in a single year. We posit that annual and km-scale variability in shoreline change can be explained by the response of different geomorphic units to time-varying

wave and ocean conditions. Ice-rich coastal bluffs and inundated tundra exhibited high retreat that is likely driven by high temperatures and wave exposure, while the stretch of shoreline with vegetated peat in front of a large breached thermokarst lake remained relatively stable. Our topographic profiles from ICESat-2 highlight three distinct shoreline types (a bluff, a small drained lake basin, and a dune in front of a large drained lake basin) that exhibit different patterns of shoreline change (both in terms of position and morphology) over the three-year study period. Analysis of altimetry-derived morphologic parameters

such as elevation and slope and small-scale features such as toppled blocks and surface ponding can provide insight on specific erosion and accretion processes that drive shoreline change. We conclude that repeat altimetry measurements from ICESat-2 and multispectral imagery provide complimentary observations that illustrate how both the position and the topography of the shoreline are changing in response to a changing Arctic.

## 1   Introduction

Decreasing sea ice extent (Overeem et al., 2011), increasing air (Serreze and Barry, 2011) and ocean (Timmermans and Labe, 2023) temperatures, and increasing storm frequency (Manson and Solomon, 2007; Irrgang et al., 2022) are driving widespread



erosion across Arctic coasts. Pan-Arctic retreat rates over the second half of the 20th century have been estimated to be -0.5 m/a, with the Alaskan Beaufort Sea Coast retreating at an elevated rate of -1.5 m/a (Lantuit et al., 2012). This erosion threatens coastal communities through damage to infrastructure and cultural sites, loss of economic opportunity, and loss of access to

traditional navigation routes and subsistence practices (e.g., Brady and Leichenko, 2020; Irrgang et al., 2022). Coastal erosion also inputs carbon and nitrogen into the ocean, impacting primary production (e.g., Terhaar et al., 2021) and the global carbon cycle (e.g., Irrgang et al., 2022). Accurate forecasts of coastal retreat rates are needed to inform carbon cycling models and coastal resilience efforts.

During the open water season when the coasts are not sheltered by sea ice, the shoreline is subjected to warm ocean tem-

peratures and mechanical energy from waves, driving ground ice thaw and erosion through thermal abrasion (Aré, 1988; Günther et al., 2013). Warm air temperatures can drive top-down ground-thawing and erosion through thermal denudation (Günther et al., 2013). Although observational studies (Nielsen et al., 2020) and models (e.g., Nielsen et al., 2022; Rolph et al., 2022) have demonstrated correspondence between environmental forcings drivers and decadal-scale retreat over regional scales (∼100s of km), studies that consider the spatial distribution of retreat rates have found them to be highly variable on

local scales (∼10s of meters) (Gibbs and Richmond, 2015; Farquharson et al., 2018; Irrgang et al., 2018; Jones et al., 2018). These findings suggest that the local response of shorelines to environmental forcings is not uniform and may depend on local shoreline characteristics.

Previous work has suggested that shoreline morphology plays a role in controlling local retreat rates. Farquharson et al. (2018) found varying shoreline change patterns across different geomorphic units in the Chukchi Sea, with permafrost bluffs

and barrier islands primarily retreating and beaches and gravel barriers showing a mixture of retreat and advance. Some observed differences are likely due to the fact that different geomorphic units are subject to a variety of erosive and accretive processes driven by time-varying ocean and atmospheric conditions. Steep coastal bluffs are predominately subjected to erosion from thermal abrasion at their base and thermal denudation above the water line (Aré, 1988; Günther et al., 2013; Irrgang et al., 2022). High rates of thermal abrasion can lead to bluff collapse events, where a cohesive block detaches from the shore-

line. These blocks can temporarily shelter the coastline from additional wave activity, but tend to disintegrate over the span of several days (Overeem et al., 2011; Barnhart et al., 2014) or weeks (Jones et al., 2018). Coastal beaches and dunes, on the other hand, are subject to erosion or accretion depending on alongshore sediment transport patterns. Observations of erosion rates and simulated wave run-up models in both temperate (Earlie et al., 2018) and Arctic (Rolph et al., 2022) environments have suggested that wave-driven retreat is sensitive to shoreline geometric attributes such as beach slope and the elevation of the

beach-cliff junction. Over time, erosion, accretion and ground thaw can drive changes in shoreline morphology which can in turn affect retreat rates. For example, surface subsidence due to ground thaw in flood-prone areas and the shrinking of beaches and barrier islands protecting the coast can both lead to accelerated thaw and erosion (Farquharson et al., 2018; Irrgang et al., 2018). Thaw slump formation (Lim et al., 2020b) and beach and barrier island growth due to sediment deposition can lead to the stabilization or advancement of the shoreline (Farquharson et al., 2018; Irrgang et al., 2018).

Quantifying the potential effect of morphology on Arctic shoreline change requires high resolution, up-to-date estimates of shoreline position, topography and geomorphic type. The increasing availability and use of high- and mid-resolution (<



10m) multispectral satellite remote sensing has facilitated the estimation of changes in shoreline position in the Arctic over large areas and long time periods (i.e. Günther et al. (2013, 2015); Farquharson et al. (2018); Irrgang et al. (2018); Jones et al. (2018)). Databases such as the Arctic Coastal Dynamics Database (Lantuit et al., 2012) and the ShoreZone project (Harper and

Morris, 2014) provide qualitative classifications of the shoreline into geomorphic units based on aerial photography and field surveys. These databases are useful for regional (Farquharson et al., 2018) and pan-Arctic (Lantuit et al., 2012; Nielsen et al., 2022) studies, but are low resolution (on the order of 10–100 km) and time-invariant, making them insufficient to examine local variations or investigate morphologic change over time. Elevation measurements from airborne lidar (e.g., Jones et al., 2013) and aerial photogrammetry (e.g., Gibbs et al., 2019; Lim et al., 2020a, b) can be used to qualitatively characterize the shoreline,

provide high-resolution estimates of shoreline position, capture short-term topographic change, and enable comparisons of retreat rates between different geomorphic units (e.g., Lim et al., 2020a) on seasonal (e.g., Gibbs et al., 2019; Lim et al., 2020a) to multi-year (e.g., Jones et al., 2013) timescales and over km-scale areas e.g., Lim et al. 2020a). Satellite-based elevation measurements enable annual and seasonal high resolution (< 5 m) coastal elevation estimates on a pan-Arctic scale, providing the potential to expand on this work and transform our understanding of Arctic shoreline morphology and change.

The Ice, Cloud and land Elevation Satellite 2 (ICESat-2) laser altimeter collects repeat cross-shore elevation profiles. ICESat-2's Advanced Topographic Laser Altimeter System (ATLAS) emits a laser pulse at 532 nm (green light), and provides elevations of individual surface-reflected photons in the ATL03 data product (Neumann et al., 2019).The laser pulse generated by ATLAS is split into 3 pairs of beams, illuminating a total of 6 ground tracks that are nominally centered around a reference ground track. Each beam within a pair is separated by 90 m across-track (i.e., perpendicular to the orbital motion of the satel-

lite), and each pair is separated by 3.3 km. ATLAS's 11 m diameter footprint (Magruder et al., 2021), 70 cm along-track sampling at full resolution (Markus et al., 2017), and cm-to-dm vertical precision (Brunt et al., 2021) allows for high resolution measurements of shoreline topography. Xie et al. (2021) and Liu et al. (2022) demonstrated the potential for using ICESat-2 altimetry to classify the shoreline by geometric unit, although these studies focused on one-time characterization. The ICESat-2 repeat-track orbit revisits the same sub-satellite ground track every 91 days, which enables measurement of annual and poten-

tially sub-annual changes in shoreline position, elevation, and shoreline morphology when the ICESat-2 mission operates in repeat-track mode. Several higher-level data products have been derived from the ATL03 photon product to reduce data volume and provide more easily interpretable elevation estimates for different applications, such as vegetation (Neuenschwander and Sheridan., 2023), land ice (Smith and the ICESat-2 Science Team., 2023), and inland water(Jasinski and the ICESat-2 Science Team., 2023). However, the resolution of these higher-level data products ($\geq$ 20 m) is not sufficient to accurately describe

complex Arctic landscapes (Michaelides et al., 2021) or measure the < 10 m changes in shoreline position that characterize much of the Arctic (Lantuit et al., 2012). New processing techniques are needed to generate coastal elevation transects from ICESat-2 photon data.

    Here, we implement a processing pipeline to generate high-resolution elevation profiles from ICESat-2 photon data and extract shoreline boundaries that can be compared with those derived from satellite multispectral imagery. We demonstrate the

utility of higher resolution ICESat-2 elevation data over the 2019–2021 open water seasons to investigate changes in cross-shore topography over time. We focus on the Beaufort Sea Coast of Alaska, where shoreline change rates are both high (averaging -22





m/a over the last decade) and variable (-48.8 m/a to 0 m/a on $\sim 10$ m length scales) (Jones et al., 2018), and multiple shoreline types (including coastal bluffs and drained lake basins, Jones et al. (2009)) are present. We quantify annual variations in sea ice cover, wave activity, and ocean and air temperatures to establish year-to-year environmental forcings on the shoreline. Next, we
derive shoreline positions and annual shoreline change from satellite multispectral imagery from Planet and evaluate temporal and spatial variations in the shoreline response to these forcings. We then analyze ICESat-2 elevation profiles and discuss the inferred shoreline type, topographic change, and potential causes of these observed shoreline changes. We also compare our ICESat-2 derived change estimates with imagery-derived change estimates. We conclude with a discussion of the advantages and challenges of characterizing shoreline structure and change with ICESat-2 altimetry and how it can be leveraged with other
datasets to better understand Arctic shoreline evolution.

## 2  Data and Methods

### 2.1  Study Area

Our study focused on an $\sim 8$ km stretch of coast to the east of Drew Point on the North Slope of Alaska (Fig. 1). This strudy area largely consists of exposed ice-rich bluffs with narrow beaches along the coast and thermokarst lakes and drained lake basins
onshore (Gibbs and Richmond, 2015). Erosion is largely driven by thermal mechanical notching of bluffs, followed by bluff collapse (Barnhart et al., 2014; Gibbs and Richmond, 2015; Jones et al., 2018). The area around Drew Point is among the most rapidly retreating locations in the Arctic, with decadal-scale shoreline change rates in excess of -10 m/a (Jones et al., 2009; Gibbs and Richmond, 2015) and evidence that retreat rates are increasing in recent years (Jones et al., 2009, 2018). Jones et al. (2018) found highly variable change in this study area, with mean shoreline change averaged over a 9 km stretch of shoreline
varying from -6.7 m to -22.0 m per open water season between 2007 and 2016 (with negative shoreline change indicating retreat). Although they found that the greatest retreat occurred in the year with most storms and warmest air temperatures, they did not find a robust relationship between spatially averaged retreat rates and open water days, storm occurrence and storm power, air temperature, permafrost temperature, or sea surface temperature over the entirety of the 10-year study period. Maximum local retreat was 2–3 times higher than spatially averaged retreat, pointing to the potential influence of local controls
(such as morphology) on shoreline change.

We divide our study area into three regions (Fig. 1b) with repeated cross-shore ICESat-2 profiles. We delineate these regions primarily based on visual analyses of 2018 optical imagery collected by CNES airbus that was accessed through Google Earth and draw from Gibbs and Richmond (2015), Jones et al. (2009), and the ShoreZone database (Harper and Morris, 2014) to describe the morphological setting of each region. ShoreZone includes a classification of variable-length segments of the
shoreline based on aerial photography taken in 2007. These classifications are based on both the composition of the shoreline and dominant erosion and accretion processes thought to be present. Given the high amount of shoreline retreat in the last decade (Jones et al., 2018), specific local features identified by ShoreZone in 2007 may have been removed or modified, but we expect the general morphologic setting to be consistent with current conditions. Region 1, the westernmost portion of the study area, is characterized by Gibbs and Richmond (2015) and Jones et al. (2009) as consisting of steep, ice-rich coastal bluffs



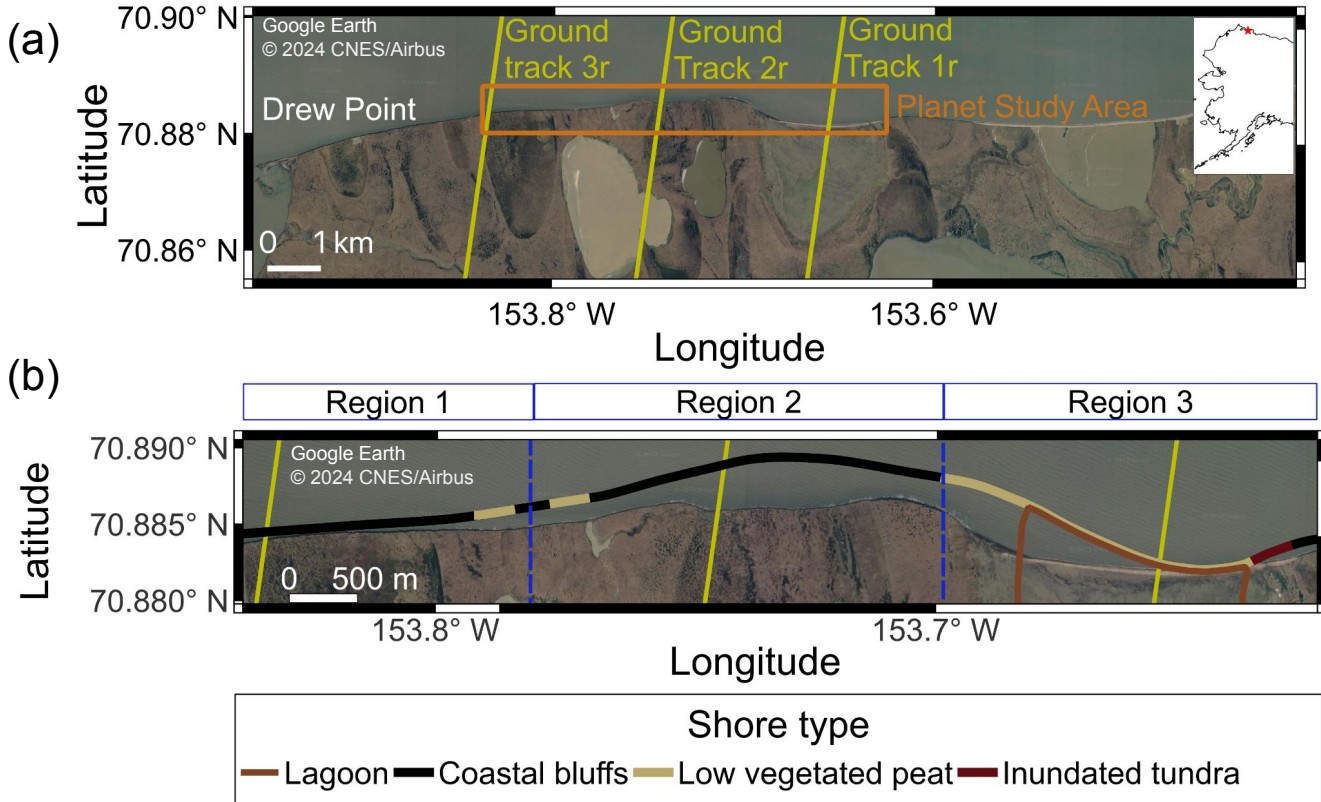

**Figure 1.** a) Overview of the Study Area on the Alaskan Beaufort Sea Coast, with the extent of the Planet imagery used and location of the ICESat-2 ground tracks indicated. b) A close-up of the study area and shore-type classifications of the 2007 shoreline from ShoreZone (Harper and Morris, 2014). We divide our study area into 3 regions based on geography and shoreline type. Background imagery from 2018 provided by CNES Airbus, courtesy of Google Earth.

(characterized as 'ground ice slumps' by ShoreZone). ShoreZone indicates the sporadic presence of low, vegetated peat-rich sediment, which are likely the remnants of old drained lake basins (Jones et al., 2009) that have since been removed by erosion. Region 2 consists of the headlands and is characterized as the same shoreline types as Region 1 (coastal bluffs and drained lake basins). Region 3 is a large breached thermokarst lake (Jones et al., 2009; Gibbs and Richmond, 2015) that is classified in ShoreZone as a large lagoon behind a narrow strip of low vegetated peat. The easternmost portion of this shoreline is classified

as a inundated tundra environment, which is characterized by low elevations, ice wedge degradation, and surface water due to ground thaw.





## 2.2 Time-varying environmental conditions

To investigate drivers of year-to-year variations in shoreline positions, we considered ocean wave conditions, air temperature, and sea ice concentration from the European Centre for Medium-Range Weather Forecasts Reanalysis v5 (ERA5) dataset
(Hersbach et al., 2020). ERA5 provides hourly estimates of atmospheric and sea ice conditions at 0.25° resolution and wave conditions at 0.5° resolution.

### 2.2.1 Sea ice and ocean waves

During the open water season, waves transport heat and mechanical energy to the base of permafrost bluffs, driving retreat via thermal mechanical abrasion (Aré, 1988). The vast majority of shoreline retreat along the Beaufort Sea Coast occurs during
the open water season, and the length of the open water season has been proposed as first-order predictor of retreat rates (Overeem et al., 2011). For each year, we estimated the duration of the open water season from the ERA5 sea daily mean sea ice concentration. Following Overeem et al. (2011), we defined the open water season as the period over which the daily mean sea ice concentration is < 15%. The open water season spans the period starting from the first day sea ice falls below 15% for at least two consecutive days to the first day sea ice remains above 15% for at least two consecutive days. We also counted the
total number of open water days (owd) spanned by each pair of Planet and ICESat-2 acquisitions, including single-day breakup events that occur outside of the open water season.

The majority of wave-driven retreat is thought to occur during storms with large waves (Barnhart et al., 2014), although time lapse imagery has shown that thermal erosion and bluff collapse can occur even under relatively calm conditions (wave heights < 0.3 m) as long as waves make contact with the base of the bluffs (Overeem et al., 2011). Thus, we considered both
the frequency of extreme wave events and total wave exposure over the open water season. We calculated the integral of the squared hourly significant wave height ($H_s$) as a proxy for cumulative wave energy. We defined the count of extreme events as the number of hours for which $H_s \geq 1.4$ m, corresponding to the upper 5% of wave heights over the 3-year study period.

### 2.2.2 Air and sea surface temperature

Ocean temperatures are an important driver of thermal mechanical erosion, as warm temperatures are required to melt frozen
sediments, which are then removed by waves. Modeling of ice-rich bluffs by Barnhart et al. (2014) showed that large retreat events only occur when retreat when ocean temperatures exceed 0 °C. Subaerial retreat, which drives the loss of sediment from the upper shoreline via permafrost thaw, is driven by warm air temperatures (Günther et al., 2013; Barnhart et al., 2014).

We aggregated daily mean air temperature from ERA5 between 1 June and 31 October of each year (following Jones et al. (2018)) and the daily mean sea surface temperature for each open water season, when the base of the shoreline is exposed to
the ocean. For both the air and sea surface temperature for each measurement period, we calculated the number of accumulated degree days of thaw (ADDT), defined as the sum of daily temperatures > 0 °C. We also recorded the mean temperature for each time interval.





## 2.3 Shoreline identification from satellite multispectral imagery

We estimated annual shoreline positions using 3 m multispectral (red, green, blue, near-IR) images from Planet Labs' Super
Dove, Dove R, and Dove Classic satellite constellations (Planet Team, 2023). We used four images, each collected near the
beginning of each open water season from 2019 to 2022 (Table 1, Fig. 2). Frequent cloud cover in 2019 necessitated the use of
imagery from 25 June, when a small amount of snow or ice remained visible near the shoreline.

For each image, we calculated the normalized differenced water index (NDWI) from the green (G) and near-IR (NIR) bands:

$$NDWI = \frac{G - NIR}{G + NIR} \tag{1}$$

We identified the NDWI threshold corresponding to the land-water boundary using Otsu's method, which determines the
threshold that divides a set of pixels into two classes such that the inter-class variance is maximized (Otsu, 1979) (see the
histograms in Fig. A1). We identified the sub-pixel land-water boundary from our NDWI images using implemented in mat-
plotlib contour in Python (Hunter, 2007). We visually identified and masked out regions where the shoreline appeared to be
misidentified due to an ambiguous land-water boundary (shaded regions in Fig. 3). In order to improve the visual agreement of
the derived shorelines with the visible shoreline in imagery and to ensure regular along-shore sampling intervals, we smoothed
each shoreline using a 30 m along-shore running mean and sampled every 10 m along the 2020 shoreline to produce our final
shoreline segments (Fig. 2). Finally, we estimated shoreline change at each segment as the cross-shore (taken in the north-south
direction) difference between successive shoreline estimates.

The estimated shorelines are subject to uncertainty from image geolocation errors, tidal- and storm-driven wave runup
influences on the land-water boundary, and errors in the threshold determined with Otsu's method. To estimate the uncertainty
in our shoreline estimates, we identified six clusters of three to six images (25 images in total) taken within a week of one
another in the early summer (mid June through late July). Modeling of coastal bluff retreat in this region by Barnhart et al.
(2014) suggests that large (> 2.5 m) erosion events are concentrated in late summer (late July-late September) and driven by
storms. Therefore, we do not expect large retreat events during early summer, such that the dominant source of change over
short periods will be transient signals such as relative sea level changes and geolocation offsets. We calculated the residual
between each shoreline position and the mean position of its cluster and pooled the residuals across all shorelines (Table A1).
We defined our shoreline estimation uncertainty to be the standard deviation of the pooled residuals.

The typical azimuthal deviation in the local shoreline from east-west average is $1.4\,° \pm 8.9°$ (Fig. A2(a)). In order to estimate
the difference in calculating shoreline relative to north-south direction rather than the local perpendicular, we estimated local
shoreline orientation at each segment based on linear interpolation of the two adjacent segments. For each annual interval, we
calculated the mean shoreline angle $\theta$ (with respect to east-west) between the two dates for each segment, and we estimated
the relative error (E) between north-south and local perpendicular change as:

$$E = 1 - cos(\theta) \tag{2}$$

We then multiplied this error by the estimated north-south change at each segment (Fig. A2 (b)) to get the final propagated
error.



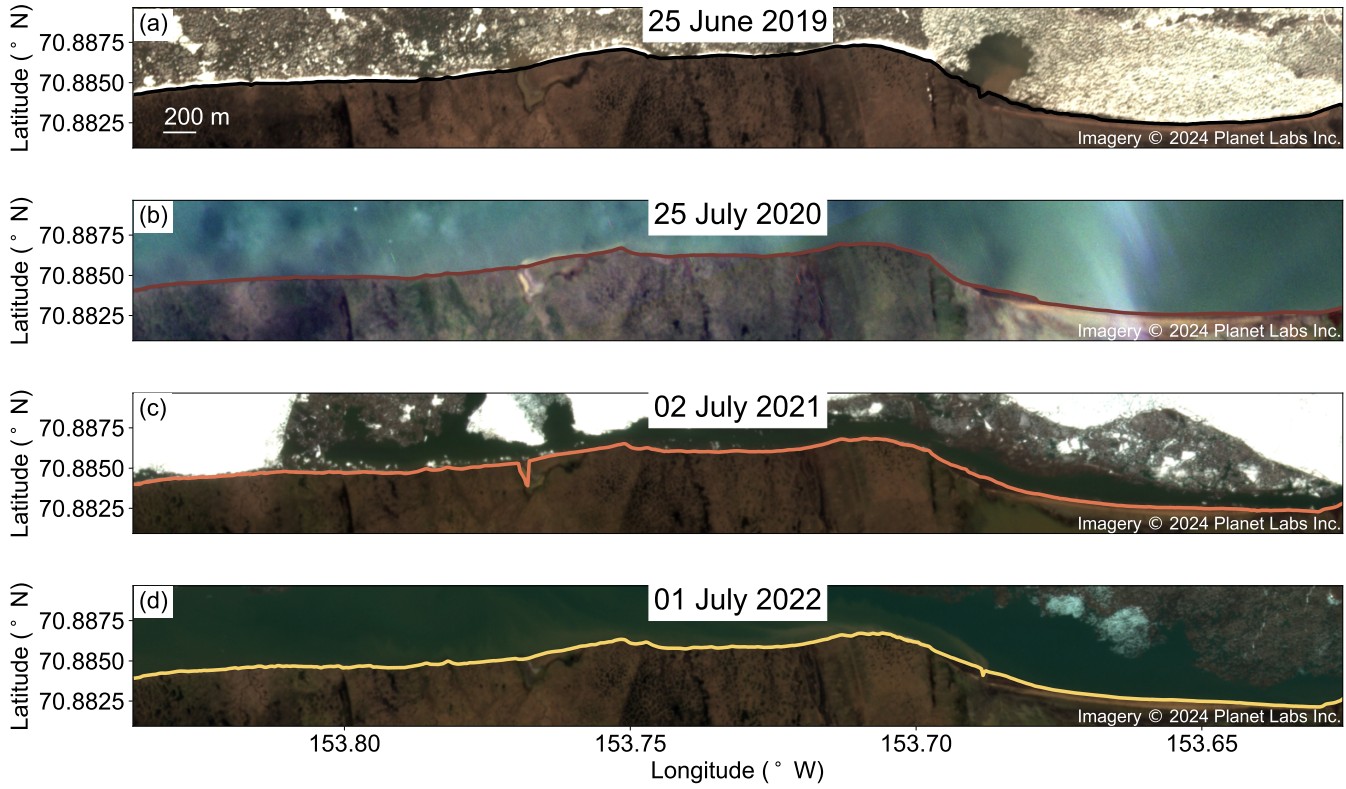

**Figure 2.** Planet Labs imagery (Planet Team, 2023) used to estimate shoreline positions. Shorelines delineated using Otsu thresholding are overlain and show good agreement with the visible land-water boundary.

**Table 1.** Observation intervals from Planet imagery and ICESat-2. The number of open water days (owd) spanned by each acquisition is listed in parenthesis.

| Interval | Planet dates | ICESat-2 dates |
|----------|--------------|----------------|
| 2019–2020 | 25 June 2019 – 25 July 2020 (154 owd) | 07 April 2019 – 04 January 2020 (138 owd) |
| 2020–2021 | 25 July 2020 – 02 July 2021 (97 owd) | 04 Janaury 2020 – 02 July 2021 (111 owd) |
| 2021–2022 | 02 July 2021 – 01 July 2022 (91 owd) | 02 July 2021 – 31 December 2021 (91 owd) |

## 2.4 Elevation profiles from ICESat-2 altimetry

We analyzed repeat ICESat-2 elevation profiles from three ground tracks (ground tracks 3r,2r, and 1r, labeled in Fig. 1 (a)) from a single reference ground track crossing our study area. Due to frequent cloud cover in the summer and fall, as well as the relatively low surface reflectivity of the snow-free tundra, the majority of observation dates with sufficient surface-reflected photons to accurately identify the surface occurred in the winter (January and April), with 2021 being the only year




with available snow-free profiles. We selected four observation dates spanning the 2019–2021 open water seasons (Table 1), resulting in 12 individual profiles.

For each profile, we used 200 m of cross-shore ATL03 photons (Neumann et al., 2023) for our analysis. Each photon from ATL03 is assigned a confidence code ranging from 0 (noise) to 4 (high confidence) to indicate how likely it is to be a

transmitted photon that was reflected from the surface (i.e. a signal photon). From each ATL03 profile, we selected all high-confidence (confidence score = 4) and medium-confidence (confidence score = 3) photons for further processing. We used the SlideRule Python Client (Shean et al., 2023) to run a customized version of the ATLAS/ICESat-2 L3A Land Ice Height (ATL06) algorithm (Smith et al., 2019), which estimates surface elevations using an iterative process of filtering and linear fitting of individual photon elevations. In this process, a linear fit is performed over fixed-length along-track segments of signal

photons, outliers are filtered out based on a specified window above and below the linear surface, and the remaining photons are re-fit. This process is iterated using a successively narrower outlier window, until either it converges or the outlier window reaches a user-specified minimum value. The reported elevation for each segment is the height of the midpoint of the final linear interpolation. SlideRule provides both uncertainties propagated from ATL03 and the RMS error between the photons used in the final fitting and the final linear fit for each segment to assess the goodness of fit.

We implemented this algorithm for 10 m long segments spaced every 2 m along-track, such that consecutive segments overlapped by 80%. A 10 m section was only considered valid if it contained at least five signal photons and if those photons were distributed over at least 1 m along-track. Bright surfaces such as surface ponds can lead to afterpulses in the ATL03 data, which appear as secondary surfaces starting ∼0.45 m below the true surface (Lu et al., 2021). The original ATL06 algorithm sets the minimum height of the outlier window to be 3 m, which leads to the inclusion of these afterpulses in the final surface

elevation estimate. To avoid including afterpulses in our analysis, our custom ATL06 processing allows for a outlier window height as narrow as 0.80 m (i.e., 0.4 m above and below the identified surface).

To estimate shoreline change from ICESat-2 data, we needed to reliably identify shoreline boundaries from derived elevation profiles. The presence of sea ice and snow in three of the ICESat-2 tracks prevents the accurate identification of a land-water boundary. Instead, we identified the boundaries of the backshore, defined here as the relatively steep region between the

beach or ocean and the onshore region. We manually identified the point corresponding to the backshore/onshore boundary (henceforth referred to as the "upper shoreline") and backshore/beach boundary (the "lower shoreline") (see Fig. A3). In order to characterize the morphology of each profile, we calculated both the backshore elevation and the backshore slope. The backshore elevation was defined as the elevation difference between the backshore/onshore boundary and the mean offshore (north of the backshore/offshore) elevation from the 02 July 2021 elevation profiles, which we used as a proxy for local sea

level (illustrated in Fig. A3). The backshore slope was estimated using a linear fit of all points between the backshore/onshore and backshore/beach boundaries. We identified the intersection between each ICESat-2 track and the corresponding imagery-derived shoreline and compared the shoreline positions and north-south retreat estimates derived from Planet and the two ICESat-2 boundaries.

To estimate the uncertainty on our ICESat-2-derived boundaries, we use the geolocation uncertainties estimated by Luthcke

et al. (2021), who co-registered ATL03 data with Arctic DEM and reported the distribution offsets between the original and





shifted locations for each beam. We consider the total uncertainty, defined as the mean offset plus one standard deviation for each beam. This total uncertainty ranges from 2.8 m to 4.8 m for individual beams. We consider the 'worst case' scenario where this error is entirely in the cross-shore direction. The satellite performs a 'yaw flip' twice every 502 days where it is re-oriented by 180 degrees (Luthcke et al., 2021). This means that the specific beam sampling a given ground track, and therefore the

associated geolocation uncertainty, alternates between repeats. We take the uncertainty in the shoreline change between two observation dates as the sum in quadrature of the uncertainties of the individual measurements, and find they range from 4.0 m to 5.9 m.

## 3 Results

### 3.1 Environmental conditions

2019 was the most extreme year in terms of all observed environmental metrics (Table 2). 2019 had the longest open water season, with sea ice breaking up sooner (late June) and re-forming later (mid-November) than in the other 2 years. It also had the highest wave energy, the most storms, higher air temperatures, and over twice as many ocean thawing degree days. By contrast, the 2021 open water season was the shortest, with sea ice breakup occurring in late July (although ERA5 suggests a one-day ice breakup on 27 June) and freeze-up occurring in late October. 2021 saw slightly warmer ocean and air temperatures

but lower wave activity (in terms of both wave energy and hours of high waves) compared to 2020.

**Table 2.** Summary of environmental variables from ERA-5 (Hersbach et al., 2020)

| Year | 2019 | 2020 | 2021 |
|---|---|---|---|
| Open water duration | 27 June – 13 November | 10 July – 26 October | 27 July – 20 October |
| Open water days | 138 | 111 | 91 |
| Cumulative wave energy ($m^2$-days) | 113.9 | 74.0 | 58.9 |
| Number of extreme wave events (hours) | 263 | 180 | 122 |
| Mean air temperature (June – October) (°C) | 3.4 | 1.6 | 1.8 |
| Mean ocean temperature (open water season) (°C) | 3.1 | 1.8 | 2.3 |
| ADDT air (°C-days) | 583.8 | 359.1 | 404.9 |
| ADDT ocean (°C-days) | 434.1 | 210.6 | 214 |

### 3.2 Imagery-derived shorelines and shoreline retreat rates

Spatially averaged shoreline in our study area was by -16.7 m/a (corresponding to retreat) between 2019 and 2021, with notable year-to-year and local variability (Table 3, Fig. 3). 2019 had the most shoreline loss, with a mean shoreline change of -24.0 m and single-segment (i.e. 10 m scale) shoreline change values as extreme as -70.1 m. 2020 experienced more moderate shoreline





change, with a mean shoreline change of -15.4 m (ranging from -41.2 m to +5.9 m). 2021 experienced similar but slightly lower rates, with a mean shoreline change of -10.6 m (-31.3 m to +5.5 m).

    Based on the uncertainty estimation described in Section 2.3, we estimated the precision of our shoreline positions estimates to be 2.2 m, corresponding to a change estimate uncertainty of 3.1 m. We define substantial shoreline change as any value that exceeds this threshold. We found that the median error associated with estimating shoreline change in the north-south

direction rather than the local perpendicular was .039 m, whereas the maximum was 4.02 m. Only 6 segments across the 3-year study period had an error larger than our 3.1 m uncertainty threshold. Thus, the difference between calculating retreat in the north-south and local cross-shore directions is minimal.

    Our shoreline change estimates for each region (Table 3, Fig. 3 (b)) indicate that there was spatial variability between regions. Region 1 showed moderately high retreat, with a mean change rate of -12.9 m/a over 3 years and single-segment year-to-year

shoreline change estimates ranging from -29.4 m to +1.5 m. Shoreline change consisted almost exclusively of retreat, with the maximum observed advancement (+1.5 m) falling below our estimated uncertainty threshold (3.1 m). We observed the greatest spatially averaged retreat in 2019 (-24.0 m), and the lowest in 2021 (-6.9 m), with 15% of valid shoreline segments (all in the western half of Region 1) not exhibiting substantial (> 3.1 m) shoreline change.

    Region 2 exhibited the largest overall retreat, with a 3-year spatially averaged mean change rate of -24.6 m/a and year-to-

year single-segment change estimates ranging from -59.3 m to -3.0 m. As with Region 1, we observed the largest amount of shoreline change in 2019 (-38.7 m), with similar and smaller change in 2020 (-17.0 m) in 2021 (-17.9 m).

    Region 3 exhibited the smallest amount of shoreline change, with a 3-year spatially averaged rate of -6.1 m/a. Region 3 was the only region in our study area where substantial shoreline advance (up to +18.5 m locally in 2019) occurred. Although 2019 was not the year with the highest spatially averaged shoreline change (-0.7 m) in Region 3, it was the most dynamic

year, with both the maximum observed local retreat (-70.1 m of shoreline change) and advance (+18.5 m of shoreline change). 24% of valid shoreline segments, concentrated almost exclusively in the easternmost third of the region, underwent significant retreat, whereas the central 62% of shoreline segments experienced substantial advance. The remaining 14% did not exhibit substantial retreat or advancement. In 2020, the mean shoreline change across the region was higher (-13.6 m), but the change at individual segments was lower in magnitude (with a local maximum of -41.2 m) and spread over a larger area (the eastern

67% of the basin). Only 3% of shoreline segments showed substantial advancement, whereas the remaining 70% experienced no significant change. Relative to 2020, retreat in 2021 was lower in magnitude (with a maximum shoreline change of -22.7 m) and present across less of the region (40%) , resulting in a lower spatially averaged shoreline change (-3.9 m). 19% of segments showed significant advancement (with a maximum of +5.5 m), whereas the remaining 41% experienced no significant change.

### 3.3   Cross-validation of ICESat-2 altimetry and Planet imagery-derived shoreline positions

Altimetry from ICESat-2 provides an independent estimate of shoreline change at multiple positions along the shoreline profile, which can be compared to Planet-derived shoreline positions for validation. We find that overall shoreline positions estimated from both datasets are consistent. Although we would expect the land-water boundary from Planet to be located north (seaward) of the backshore boundaries identified by ICESat-2, we note that it consistently falls between the upper and lower



**Table 3.** Estimated shoreline change between each successive image observation date in each of the three regions identified in Figure 3, as well as across the whole study area. The mean and range are listed.

| Year | Region 1 | Region 2 | Region 3 | Total |
|---|---|---|---|---|
| 2019 | -18.6 m (-29.4m – -7.15 m) | -38.7 m (-59.3 m – -16.8 m) | -0.7 m (-70.1 m – +18.5 m) | -24.0 m (-70.1 m – +18.5 m) |
| 2020 | -13.0 m (-21.6 m – -3.4 m) | -17.0 m (-27.7 m – -3.0 m) | -13.6 (-41.2 m – +5.9 m) | -15.4 m (-41.2 m – + 5.9 m) |
| 2021 | -7.1 m (-14.1 m – +1.5 m) | -17.9 m (-30.6 m – -5.6 m) | -3.9 (-22.7 m – +5.5 m) | -10.6 m (-30.6 m – +5.5 m) |
| 2019–2021 | -12.9 m (-29.4 m – +1.5 m) | -24.6 m (-59.3 m – -3.0 m) | -6.1 m (-70.1 m – +18.5 m) | -16.7 m (-70.1 m – +18.5 m) |

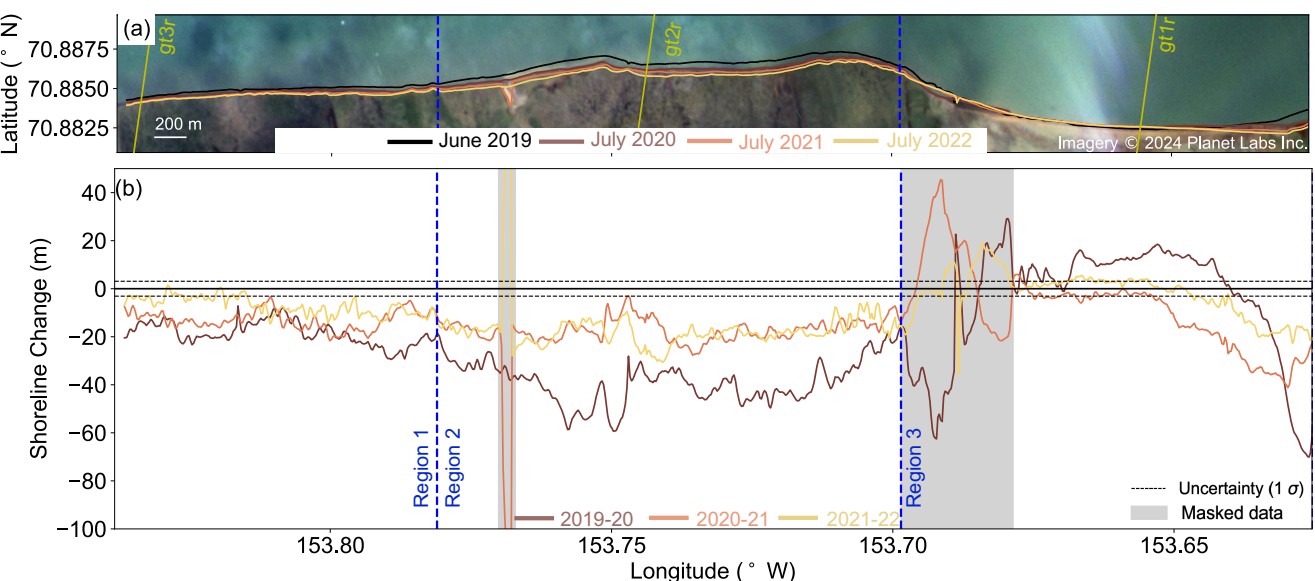

**Figure 3.** (a) Shorelines derived from Planet images, overlain on imagery from 07/25/2020. The locations of the three ICESat-2 ground track are also shown. (b) North-south shoreline change calculated between successive years. Areas where there was an ambigous land/water boundary are masked out (shaded areas) and excluded from analysis. The boundaries of the three region shown in Fig. 1 are shown as dashed lines.

shoreline boundaries when compared to both snow-on and snow-off ICESat-2 observations (Fig. 4). This could be explained
by a consistent landward bias in our NDWI thresholding technique (Section 2.3), geolocation offsets between Planet imagery and ICESat-2 tracks, or differences introduced by snow cover and variations in the local water level. Although observations from ICESat-2 and Planet span different time intervals (Table 1, Fig. 4), the number of open water days spanned by both are similar, allowing for a direct comparison between our imagery and altimetry-defined shoreline change estimates. We perform





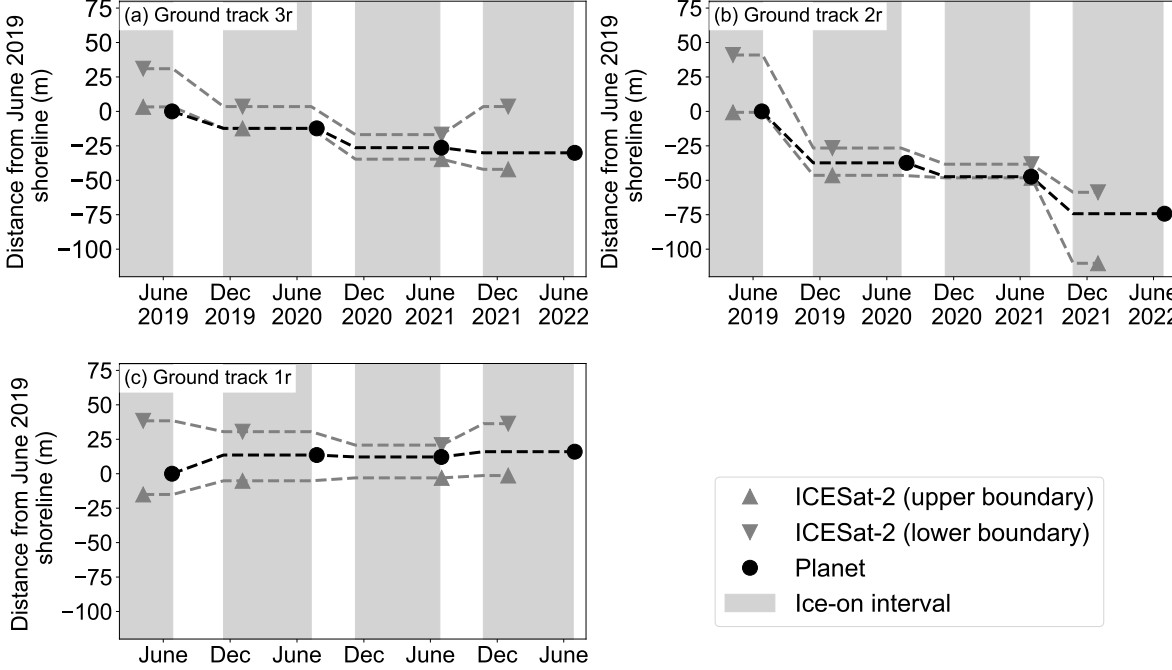

**Figure 4.** Time series of the position of upper and lower boundaries and coastal boundaries from ICESat-2 and the land-water boundary derived from Planet. Positions are given with respect to the location of Planet-derived shoreline from 25 June 2019. Ice-on intervals are shaded.

an orthogonal distance linear regression and find a high correlation between the Planet-derived change estimates and changes in the ICESat-2-derived upper boundary ($r^2$ = 0.79, p = .0007) and moderate correlation with the ICESat-2-derived lower boundary ($r^2$ = 0.53, p = .007) (Fig 5 (a)).

### 3.4 Topographic change from satellite altimetry

Our SlideRule-derived elevation profiles (Fig. 6 (b), Fig. 6 (d) and Fig. 6 (f)) fit the ATL03 photon data well, with RMS errors ranging from 0.04 m to 1.3 m and a mean RMS error of 0.15 m (Fig. A4, Fig. A5, Fig. A6). The propagated vertical uncertainty ranges from 0.0024 to 0.27 m, with a mean uncertainty of 0.029 m. We now discuss the time evolution of the shoreline horizontal position (Table A2), backshore elevation, and backshore slope (Table 4) at each ICESat-2 ground track.

Ground track 3r in Region 1 (Fig. 6 (a), Fig. 6 (b), Fig. 4(a)) shows a coastal bluff (such as the one shown in Fig. A7(a)) that undergoes retreat with little change in morphology. The upper and lower shoreline boundaries show consistent retreat in 2019 and 2020, with north-south change at each boundary ranging from -15.3 m to -27.5 m. During these 2 years, we observed a steepening of the backshore slope from 16% in 2019 to 29% in 2020 and 30% in 2021. In July 2021, we note a cluster of photons that is ∼1 m high and ∼5 m across at the base of the bluff (Fig. 7(a)) that may correspond to toppled bluff material. Between 2021 and 2022, we observed slight retreat of the upper boundary (-7.4 m of shoreline change) and advancement of





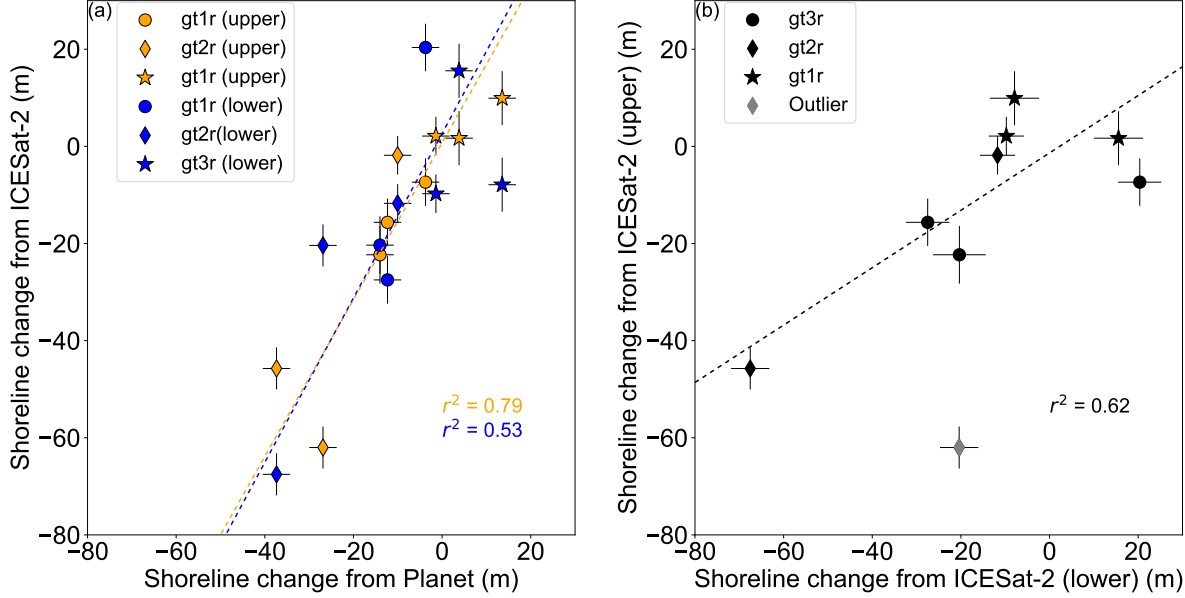

**Figure 5.** a) Comparison between shoreline change estimates from Planet and shoreline change estimates from the upper (orange) and lower (blue) boundaries derived from ICESat-2. b) Comparison between the measured change between each open water season across the upper and lower boundaries derived from ICESat-2. The coefficient of determination excluding the outlier drained lake measurement (grey) is reported. Linear fits were estimated using orthogonal distance regression.

the lower boundary (+20.4 m), resulting in a reduction in the backshore slope to 11%. In December 2021, we again note a ∼1 m high cluster of photons at the base of the bluff, that gently slopes down towards the mean offshore height over ∼20 m (Fig.

7(a)). The backshore elevation remains stable (ranging from 4.89 m to 5.02 m) over the 3 year study period.

Ground track 2r (Fig. 6 (c), Fig. 6 (d) Fig. 4(b)) in Region 2 passes over the remnant basin of a small (∼150 m in diameter) lake that was breached and drained as a result of shoreline retreat prior to our study. This profile undergoes large changes in both the shoreline position and morphology. Aerial imagery from ShoreZone (Fig. A7(b)) showed the lake pre-drainage, and the lake basin was clearly visible in Airbus imagery from Google Earth (Fig. 6(c)) and in our 2019 ICESat-2 profile (Fig. 6(d)).

Between 2019 and 2020, -47.5 m of upper-shoreline change and -67.5 m of lower shoreline change occured, corresponding to the loss of about half of the basin. This resulted in a 0.40 m drop in the backshore height (from 1.33 m to 0.93 m) and an increase in the backshore slope from 2.5% to 4.3%. By contrast, between early 2020 and early 2021 there was almost no change in the upper shoreline position (-1.8 m) and only moderate change in the lower shoreline position (-11.7 m), resulting in a further increase in the backshore slope to 6.7%. We also observe a slight lowering (by 0.23 m) of the backshore height.

The photon distribution over the lake basin in July 2021 is concentrated in a narrow height band, with a secondary reflection below the surface (Fig. 7(b)). This feature is indicative of an afterpulse due to surface ponding (Lu et al., 2021), suggesting





**Figure 6.** (a), (c), (e):Detailed view of the ICESat-2 sampling location from 2018 Google Earth Imagery for (a) ground track 3r; (c) ground track 2r and (e) ground track 1r . The location of the Planet shoreline and corresponding ICESat-2 pass for each year is shown. (b), (d), (f): ATL03 photon clouds from each ICESat-2 pass, with the SlideRulederived (ATL06-SR) elevation profile overlain for (b) ground track 3r; (d) ground track 2r; anf (f) ground track 1r. The estimated location of the upper and lower shoreline boundary for each date is marked, along with the north-south locations of the Planet-derived shorelines. The colors follow the same legend as (a), (c), and (e).





**Table 4.** Onshore heights and slopes derived from the identified upper and lower boundaries for each ICESat-2 observation.

| Track | Date | Backshore elevation (m) | Backshore slope (%) |
|---|---|---|---|
| Ground Track 3r | 07 April 2019 | 5.02 | 16 |
| (Region 1) | 04 January 2020 | 4.87 | 29 |
| | 02 July 2021 | 4.89 | 30 |
| | 31 December 2021 | 4.96 | 11 |
| Ground Track 2r | 07 April 2019 | 1.33 | 2.5 |
| (Region 2) | 04 January 2020 | 0.93 | 4.3 |
| | 02 July 2021 | 0.70 | 6.7 |
| | 31 December 2021 | 2.04 | 3.9 |
| Ground Track 1r | 07 April 2019 | 1.89 | 3.7 |
| (Region 3) | 04 January 2020 | 1.67 | 6.7 |
| | 02 July 2021 | 1.64 | 7.7 |
| | 31 December 2021 | 1.64 | 4.4 |

the presence of shallow water. The rest of the lake basin was eroded between July and December 2021, resulting in -62.0 m of change of the upper shoreline and -30.1 m of change of the lower shoreline. This necessitated the definition of a new, substantially higher (2.04 m) backshore boundary and resulted in a shallowing of the backshore slope from 6.7% to 3.9%.

325   Ground track 1r (Fig. 6 (e), 6 (f), Fig. 4(c)) in Region 3 passes over a 1.71 m high dune in front of a large ($\sim$ 2.6 km across) breached thermokarst lake (shown in more detail in Fig. A7(c)) that shows little shoreline change relative to other regions. We observed between +1.7 m and +9.9 m of advance in the upper boundary each year, while the lower boundary retreated in 2019 (-7.9 m) and 2020 (-9.8 m) and then advanced in 2021 (+17.5 m). We note a 0.22 m drop in the backshore elevation (from 1.89 m to 1.67 m) between 2019 and 2020, after which the elevation remains stable from 2020 to late 2021 (1.64 m to 1.67 m). The
backshore slope steepens between 2019 and 2021 from 3.7% to 7.7 %, and then relaxes to 4.4% in late 2021.

## 4   Discussion

### 4.1   Spatiotemporal variability in shoreline change in the context of previous observations

Our estimates of spatially averaged (regional) mean annual shoreline change (-10.7 m/a to -29.0 m/a) across our study region are higher than long-term historical estimates and similar to recent observations. Gibbs and Richmond (2015) estimated a
regional mean of -6.3 m/a and a local maximum of -18.6 m/a between Drew Point and Cape Halket between 1947 and 2002. Jones et al. (2009) estimated shoreline change across this region over multiple time intervals and found -6.8 m/a of change between 1955 and 1979, -8.7 m/a between 1979 and 2002, and -13.6 m/a between 2002 and 2007. A follow-up study by Jones





**Figure 7.** ICESat-2 ATL03 photon data show small-scale features, including (a) returns over what may be a toppled block material at the base of the bluff at ground track 3r and (b) Surface ponding in the drained lake basin at ground track 2r.





et al. (2018) estimating shoreline change over a 9 km region covering our study area found a 10-year mean shoreline change rate of -17.2 m/a between 2007 and 2016. Taken together, these studies suggest a sustained acceleration in retreat rates over
the past few decades. Our observed retreat rates are consistent with this increasing trend. However, we note that our elevated estimates of erosion may be in part due to the short time period of our observations, as short-term estimates of shoreline change tend to be more variable than long-term estimates (Sadler and Jerolmack, 2015). In addition to estimating a 10-year mean, Jones et al. (2018) reported regional year-to-year rates, which we use to put our observations in the context of recent short-term observations. Our mean retreat rates in 2020 and 2021 fall within the range of year-to-year rates observed by Jones
et al. (2018) (-6.7 m/a to -22.6 m/a), whereas our mean retreat in 2019 (-29.0 m) is exceeds that range.

Previous observations suggest that the spatial distribution of change rates across our study area has been variable over time. Gibbs and Richmond (2015) found the highest rates of shoreline change in Region 3 (-10 m/a to -18 m/a), intermediate rates in Region 2 (-8 m/a to -12 m/a), and slightly lower rates in Region 1 (-6 m/a to -12 m/a) between 1947 and 2002. Jones et al. (2009) found a similar pattern between 1955 and 1979, with the shoreline change in Region 3 occuring at rates in excess of -10
m/a and shoreline change in most of Regions 1 and 2 occuring at rates between -5 m/a and -10 m/a. However, they found that shoreline change rates in Region 2 increased to be in excess of -10 m/a between 1979 and 2002, and change rates in Region 1 followed suit between 2002 and 2007, such that change rates across the region were relatively uniform (averaging -18 m/a) between 2002 and 2007.

In contrast, our observed shoreline change rates between the three regions between 2019 and 2021 are not uniform. Although
the relatively high rates of retreat we observed in Region 2 compared to Region 1 is consistent with Gibbs et al. (2019) and Jones et al. (2009), the stability of the central portion of Region 3 that we observed with both ICESat-2 altimetry and Planet imagery appears to be a recent development. Wang et al. (2022) sampled a transect across the central portion of Region 3 from satellite imagery-derived shorelines from 1974, 1985, 1992, 2001, 2009, and 2017 and estimated a mean shoreline change rate over the full study period of -55.9 m/a. When considering the rates between successive shorelines, Wang et al. (2022) found
that 2009–2017 was the only time interval during which there was no observed shoreline change, suggesting that stabilization occurred in this time frame after over 4 decades of retreat.

## 4.2  Drivers of spatiotemporal variability in imagery-derived shorline change

Jones et al. (2018) found that the environmental drivers of year-to-year variability on the Beaufort Sea Coast are not well-defined, but they did observe high retreat in years with extreme weather, which is consistent with our findings of both high
retreat and extreme environmental conditions in 2019. Compared to 2020 and 2021, 2019 had a long open water season and elevated air and ocean temperatures (Table 2), all of which likely contributed to the elevated spatially averaged retreat across the study region.

Although there is correspondence between mean year-to-year shoreline change and year-to-year variations in wave and temperature conditions, the response of the shoreline to these time-varying conditions is not spatially uniform (Fig. 3 (b)). In
2019, Region 1 and Region 2 both experienced high retreat, while Region 3 underwent high retreat across parts of the shorelines and moderate advancement across others. Whereas we do not expect air and ocean temperatures to vary greatly spatially over





the study region, local variations in wave energy due to shoreline orientation, position, and morphology may contribute to the observed spatial variability. For indented coastlines such as our study area, wave energy is expected to be concentrated towards the headlands due to wave refraction, such that coastlines with uniform composition will straighten over time (Van Rijn, 2011).

This provides a potential explanation for increased retreat in Region 2, particularly in 2019 when wave action was high. 2021 saw the lowest amount of wave energy and fewest storms (Table 2), and the concentration of wave energy in the headlands of Region 2 would result in particularly low wave energy in the hinterlands of Region 1 and Region 3. We note that while Region 1 and Region 2 both consist primarily of ice-rich coastal bluffs, the shoreline in Region 3 is characterized by ShoreZone primarily as low-lying peat (Fig. 1 (b)) that is expected to be subject to low incident wave energy (Harper and Morris, 2014).

The eastern edge of Region 3 is characterized as inundated tundra, which refers to areas characterized by high thaw subsidence and surface ponding (Harper and Morris, 2014). Based on these characterizations and our observed patterns shoreline change, we posit that shoreline change in Region 1 and Region 2 is sensitive to year-to-year variations in wave energy and ocean and air temperatures, while the western and central portion of Region 3 are not. The eastern portion of Region 3 (inundated tundra) is likely subject to ground thaw, high wave energy and flooding, resulting in high retreat in response to high wave activity and

high temperatures. The advancement observed in Region 3 may be driven by along-shore transport of material lost from the surrounding regions.

### 4.3 Drivers of morphologic change observed from ICESat-2 altimetry

The elevation profiles from ICESat-2 data provide additional information on topographic and morphological change at specific ground tracks (Table 4), which can provide insight on specific erosion and accretion processes that drive local shoreline change.

Specifically, we consider changes in backshore elevation, backshore slope, and relative change between the upper and lower shoreline boundaries.

Based on previous studies in the Beaufort Sea Coast area (e.g., Overeem et al., 2011; Barnhart et al., 2014), we infer that the coastal bluff shown in ground track 3r in Region 1 (Fig. 6 (b)) likely retreats primarily through the formation of thermal erosional niches, followed by bluff collapse. As the shoreline retreats, we observed no major morphological change;

the backshore slope remained > 10% with a stable (4.87 m to 5.02 m) backshore elevation. However, there were year-to-year fluctuations in the backshore slope due to differences in change rates at the upper and lower boundaries of the backshore. In particular, the relatively low slope (11%) in December 2021 is driven by a retreat of the upper boundary (-7.4 m of shoreline change) coupled with advance of the lower boundary (+20.4 m of shoreline change) (Table A2). Based on the ICESat-2 ATL03 photon data from July and December 2021 (Fig. 7(a)), we hypothesize that this reduction in slope is due to the accumulation

of collapsed bluff material at the base of the cliff that is not removed by the end of the 2021 open water season. The low retreat derived from altimetry in 2021 is consistent with the low retreat estimated from Planet imagery in the broader Region 1 (which has a spatially averaged mean shoreline change of -7 m) and is possibly due to relatively low wave energy and fewer storms in the 2021 open water season(Table 2). It may be that that multiple locations along the shoreline in Region 1 observed in imagery were sheltered by uneroded toppled bluff material similar to what we observe at ground track 3r.





The drained lake basin captured by ground track 2r in Region 2 (Fig. 6 (d)) is an example of a more complex feature, which displays variable retreat as the shoreline migrates from the lake basin boundary to the bluff on the southern edge. The transition in shoreline morphology is reflected in an increase in backshore elevation from 0.70 m to 2.04 m and moderate fluctuations in the backshore slope (2.5 % to 6.7 % ; Table 4). The changes in slope between 2019 and 2020 and between early and late 2021 were driven by high and variable (-30.0 m to - 67.5 m) change in both the upper and lower boundaries, while the steepening of

the slope between 2020 and 2021 was driven by retreat of the lower shoreline (-11.75 m of shoreline change) and little change in the upper shoreline (-1.8 m).

In order to assess whether the presence of this lake basin impacts local retreat rates, we compared the mean Planet-derived shoreline change across the drained lake basin (which spans 15 segments over 150 m) to the distribution of Planet-derived shoreline change across the entirety of Region 2. In 2019, the mean shoreline change across the lake basin (-35.3 m) is similar

to retreat in the Region 2 overall, as it falls in the 35th percentile of observed change. However, in 2020 shoreline change across the basin (-10.5 m) fell into the 7th percentile of observed headlands change, and in 2021 it was in 90th percentile (-23.6 m). This suggests that the presence of the drained lake basin may have contributed to differential erosion during 2020 and 2021 relative to Region 2 overall. Differential erosion rates in drained lake basins have been observed before, with Jones et al. (2009) observing generally higher retreat rates of recently drained lake basins in this region between 1955 and 2002 relative to other

land types (although this trend did not hold between 2002 and 2007). They suggested that the low elevation and presence of thawed sediments likely makes drained lake basins more susceptible to erosion than coastal bluffs. Our observation of surface water in July 2021 (Fig. 7 (b)) is consistent with this interpretation, as it would help keep the underlying sediments in this basin thawed.

Based on our observations and the theory presented in Jones (2009), we can infer the processes driving the 3-year evolution

of the drained lake basin. In 2019, wave-driven erosion of the seaward side of the drained lake basin exposed low-lying thawed sediments. In 2020, debris from this large erosion event may have sheltered the remainder of this basin, as evidenced by the retreat at the lower boundary and stability of the upper boundary. Once this debris was removed by waves, the low backshore elevation (0.70 m) would have left the basin susceptible to overtopping by high waves, and the thawed sediments would have low mechanical strength, making the basin susceptible to rapid erosion as observed between early and late 2021. The steeper

backshore slope in 2019 compared to 2021 may have also contributed to increased wave run-up (Earlie et al., 2018) and therefore increased mechanical erosion.

Both the upper and lower shoreline positions at ground track 1r in Region 3 (Fig. 6 (f), Fig. 4 (c)) are relatively stable, whereas the other ground tracks exhibit high retreat. We see moderate advance (+9.9 m of shoreline change) and a drop in elevation (-0.22 m) of the backshore boundary in 2019, but very little change in either the position (+1.7 m to +2.1 m, which

falls within our estimated uncertainty) or elevation (which ranges from 1.64 m to 1.67 m) over the next 2 years. Fluctuations in the slope between observation dates are driven primarily by changes in the lower boundary, which is consistent with sediment deposition and removal at the beach in front of the lagoon or changes in snow cover.

In order to understand the relative rates of retreat at the upper and lower boundaries of the shoreline, we perform an orthogonal distance regression between the annual change at both boundaries (Fig. 5(b)). We exclude change for the drained lake





basin between July and October 2021, where the collapse of the remaining lake basin necessitated the definition of new upper shoreline boundary. We find a moderate correlation between upper and lower shoreline boundary change ($r^2$ = 0.62, p = .009) and a slope of 0.59 in the linear fit. Thus, both boundaries exhibit comparable amounts of change, but the lower shoreline (the interface between the backshore, beach, and ocean) is more dynamic.

### 4.4  Potential and challenges using ICESat-2 for shoreline characterization

ICESat-2's geolocated photon product resolves coastal topography with a high level of detail, capturing abrupt elevation changes over horizontal distances of 5 m or less (e.g Fig. 6 (b)). We observe process-scale features such as toppled blocks and surface ponds in snow-free ICESat-2 profiles (Fig 7). We find that the simplified elevation profiles derived via SlideRule can adequately capture the shoreline for assessing shoreline evolution, although small-scale features such as the toppled block on ground track 3r (Fig. 7 (a)) and surface ponding on ground track 2r (Fig. 7 (b)) are lost. ICESat-2 profiles allow for the
identification of multiple shoreline boundaries, which can be compared against imagery-derived shoreline positions (Fig. 4, Fig. 5 (a)). Furthermore, analysis of morphologic parameters such as backshore height and backshore slope can provide insight on specific erosional and accretional processes, such as the collapse of the small drained lake and the presence of toppled blocks at steep coastal bluffs.

The highest RMS misfit between the SlideRule-derived elevation profiles and underlying photons occur in areas with steep
slopes and abrupt elevation changes (Fig. A4, Fig. A5, Fig. A6). This is particularly apparent for ground track 3r (Fig. A4), where the upper shoreline boundary inferred from SlideRule often appears 2 m to 5 m southward of the boundary that would be visually inferred from ATL03. The 10 m section length we used in SlideRule is likely too coarse to adequately capture abrupt elevation changes, but a shorter segment length would have relied on fewer photons per segment and therefore be more sensitive to variations in along-track photon density. An adaptive approach based on local topography and photon density may
be able to capture steep and complex features more accurately.

Given the seasonal revisit time of ICESat-2 over our study region, the high frequency of cloud cover in the summer and fall, and the low reflectivity of the snow-free tundra, the majority of usable ICESat-2 data occurs in winter and spring, when snow and sea ice are present. Snow cover will impact ICESat-2-based shoreline position estimates, particularly the location of the lower shoreline boundary, which will in turn impact our slope estimates. Although we expect the magnitude of snowcover-
induced changes to be small relative to the large erosion events observed at the ground track 3r (the coastal bluff) and the ground track 2r (the small drained lake), it may be the dominant source of change in areas with lower rates of change, such as the ground track 1r (the large breached lake). Future work comparing snow-on ICESat-2 tracks to snow-off ICESat-2 tracks and snow-off digital elevation models could help quantify snow distribution at the shoreline and its impact on shoreline position estimates. There is also up to 15 m of horizontal offset between repeat ICESat-2 profiles (Fig. 6(a), Fig. 6(c), Fig. 6(e)),
meaning that differences in shoreline position may in part be due to different sampling locations.



## 5 Conclusions

We used multispectral imagery from Planet and altimetry data from ICESat-2 to highlight spatiotemporal variability in changes in the position and topography of the shoreline along a dynamic section of the Beaufort Sea Coast near Drew Point, AK USA. We found annual km-scale variability in shoreline change that reflects the response of distinct geomorphic units to time-varying wave and temperature conditions and small-scale variability (∼10s of m) that may be influenced by local shoreline morphology. We used elevation profiles from ICESat-2 altimetry to track changes in the position, elevation, and cross-shore slope of the shoreline at three ground tracks. We found that each ground track samples a distinct shoreline type that is subject to different mechanisms of change. At ground track 3r (Region 1) and ground track 1r (Region 3), we observed changes in shoreline position that are consistent with adjacent imagery-derived estimates, and analysis of changes in morphological parameters (namely elevation and and slope) helped to illustrate specific processes (such as rapid bluff retreat, sheltering of bluffs by collapse bluff material, and sediment accumulation/deposition) that drive the observed shoreline change. Ground track 2r in Region 2 illustrates how a small-scale feature (in this case a drained lake basin) can be subject to different processes than the surrounding area, leading to locally variable retreat.

Overall, we found that annual retreat rates from both datasets are comparable to previous estimates of shoreline change over the last decade and that current spatial patterns of retreat differ from long-term trends, particularly in Region 3. Planet imagery and ICESat-2 altimetry provided complementary shoreline measurements, with the two datasets producing similar estimates of shoreline position change. Multispectral imagery can provide regular scene-wide estimates of shoreline change that can contextualize ICESat-2 topographic transects with the surrounding shoreline. Altimetry data from ICESat-2 provides cross-shore estimates of topographic change that allow us simultaneously track changes in shoreline position and morphology; this vertical dimension critically provides additional insight into the geomorphic processes driving shoreline change. Small features that are visible in the ATL03 photon data can also aid in the interpretation of short-term processes. Additional datasets such as aerial photography and shoreline classifications from databases such as ShoreZone can also aid in the interpretation of both satellite altimetry and satellite imagery. Retreat rates derived from altimetry and satellite imagery can also be used for cross-validation, and the ability to estimate both the upper and lower shoreline boundary from altimetry allows us to better interpret our shoreline estimates derived from multispectral imagery.

The regular revisits and dense track sampling of ICESat-2 provides estimates of retreat rates and geometric properties such as elevation and slope over large areas and multiple years. However, bulk analysis of these changes requires a method for systematic processing of ATL03 data to generate simplified elevation profiles and extract relevant parameters. The processing workflow presented here generates elevation profiles that show good agreement with ATL03, but could be improved upon to better capture abrupt elevation changes and small-scale features. Future work should also develop techniques to estimate contributions from snow and topographic offset.



*Data availability.* The ICESat-2 ATL03 geolocated photon product is available at the National Snow and Ice Data Center (NSIDC): https://doi.org/10.5067/ATLAS/ATL03.006. ERA-5 data are available at the Copernicus Data Store: https://doi.org/10.24381/cds.adbb2d47. Aerial photography and shoreline classifications are available at the NOAA Fisheries Alaska ShoreZone Website:

https://alaskafisheries.noaa.gov/mapping/sz. Multispectal imagery was provided by Planet Labs under the NASA Commercial Smallsat Data Acquisition Program (CSDA). Derived datasets including shoreline positions and elevation profiles are available at: https://zenodo.org/records/11095272

*Author contributions.* MBB was responsible for the design, data analysis, visualization, and preparation of the manuscript. AAB worked with the lead author to conceptualize the research that went into the manuscript, discussed all aspects of the investigation and analysis with the lead author throughout the project, and contributed to review and editing of all drafts of the manuscript. All other authors were involved

in the conceptualization of the project and reviewed and edited all drafts of the manuscript.

*Competing interests.* The authors declare that there are no competing interests.

*Acknowledgements.* This work was supported by NASA grant no. 80NSSC24K0019 (MB), 80NSSC22K1105 (RJM), and 80NSSC21K0912 (MRS). We thank the Scripps glaciology group for advice on analyzing and visualizing ICESat-2 data. The Scientific colour map lajolla (Crameri, 2023) is used in this study to prevent visual distortion of the data and exclusion of readers with colour-vision deficiencies (Crameri

et al., 2020).



**Appendix A**

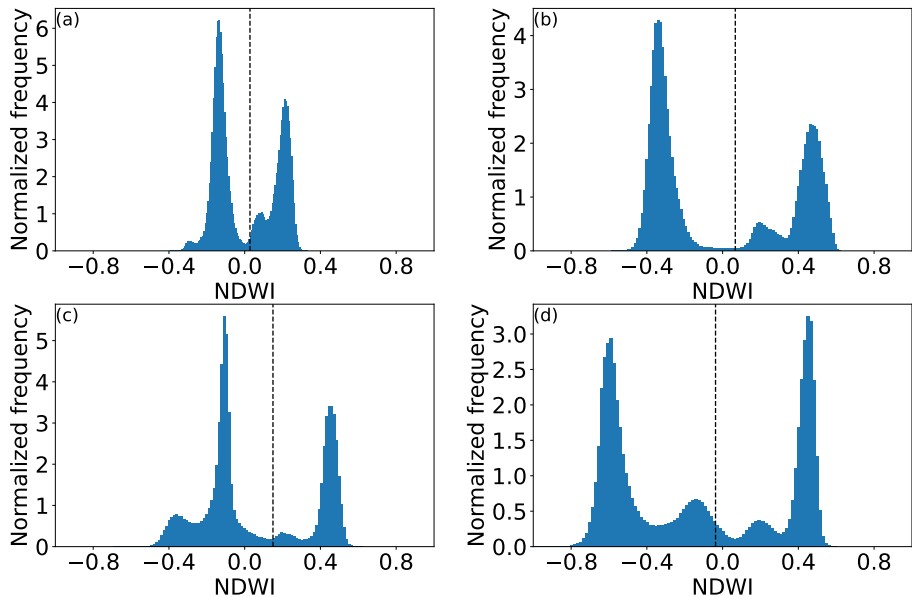

**Figure A1.** Normalized distribution of scene-wide NDWI values for Planet images collected on (a) 25 June 2019, (b) 25 July 2020, (c) 02 July 2021, and (d) 01 July 2022. The threshold used for identifying the land-water boundary, estimated using Otsu's method, is shown as a dashed vertical line.

**Table A1.** The dates of all images used for the cluster uncertainty analysis described in Section 2.2, along with the standard deviation of the residuals derived from each cluster. The total number of images from each date is listed in parenthesis

| Cluster | Dates | Standard deviation of residuals (m) |
|---|---|---|
| 1 | 26 June 2020 (1), 27 June 2020 (1), 29 June 2020 (2) | 1.9 |
| 2 | 04 July 2020 (1), 07 July 2020 (4), 07 10 2020 (1) | 2.3 |
| 3 | 25 July 2020 (3) | 2.9 |
| 4 | 16 June 2021 (2), 19 June 2021 (1) | 1.9 |
| 5 | 02 July 2021 (1), 04 July 2021 (2) | 2.5 |
| 6 | 13 July 2021 (1), 14 July 2021 (3) | 2.0 |



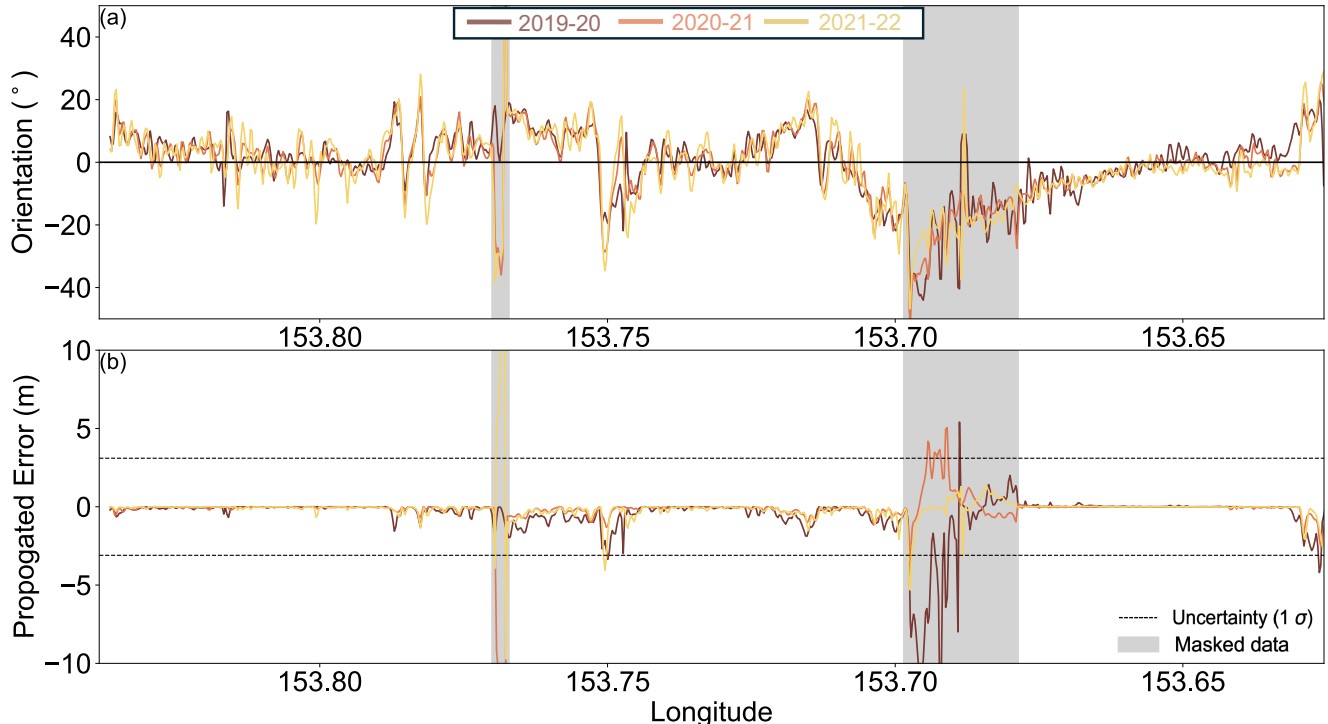

**Figure A2.** (a) Local orientation of the shoreline averaged between consecutive years, measured clockwise from north. (b) The relative error calculated from Equation 2 based on these orientations scaled by the measured shoreline change (Fig. 3b). The majority of this error falls well below our uncertainty threshold estimated in Section 2.3 (plotted as a dashed line).

**Table A2.** North-South change in the upper and lower boundaries estimated from ICESat-2 and the land-water boundaries estimated from Planet at each sampled ICESat-2 location

| Track | Boundary | 2019 | 2020 | 2021 |
|---|---|---|---|---|
| Ground Track 3r | upper boundary (Region 1) | -15.6 m ($\pm$ 4.9 m) | -22.3m ($\pm$ 5.9 m) | -7.4 m ($\pm$ 4.9 m) |
| | lower boundary | -27.5 m ($\pm$ 4.9 m) | -20.3 m ($\pm$ 5.9 m) | 20.4 m ($\pm$ 4.9 m) |
| | land-water boundary (Planet) | -12.3 m ($\pm$ 3.1 m) | -14.0 m ($\pm$ 3.1 m) | -3.7 m ($\pm$ 3.1 m) |
| Ground track 2r | upper boundary | -45.7 m ($\pm$ 4.3 m) | -1.8 m ($\pm$ 4.0 m) | -62.0 m ($\pm$ 4.3 m) |
| (Region 2) | lower boundary | -67.5 m ($\pm$ 4.3 m) | -11.7 m ($\pm$ 4.0) | -30.1 ($\pm$ 4.3 m) |
| | land-water boundary (Planet) | -37.4 m ($\pm$ 3.1 m) | -10.0 m ($\pm$ 3.1 m) | -26.9 m ($\pm$ 3.1 m) |
| Ground track 1r | upper boundary | 9.9 m ($\pm$ 5.6 m) | 2.1 m ($\pm$ 4.0 m) | 1.7 m ($\pm$ 5.6 m) |
| (Region 3) | lower boundary | -7.9 m ($\pm$ 5.6 m) | -9.8 m ($\pm$ 4.0 m) | 17.5 m ($\pm$ 5.6 m) |
| | land-water boundary (Planet) | 13.6 m ($\pm$ 3.1 m) | -1.4 m ($\pm$ 3.1 m) | 3.8 m ($\pm$ 3.1 m) |



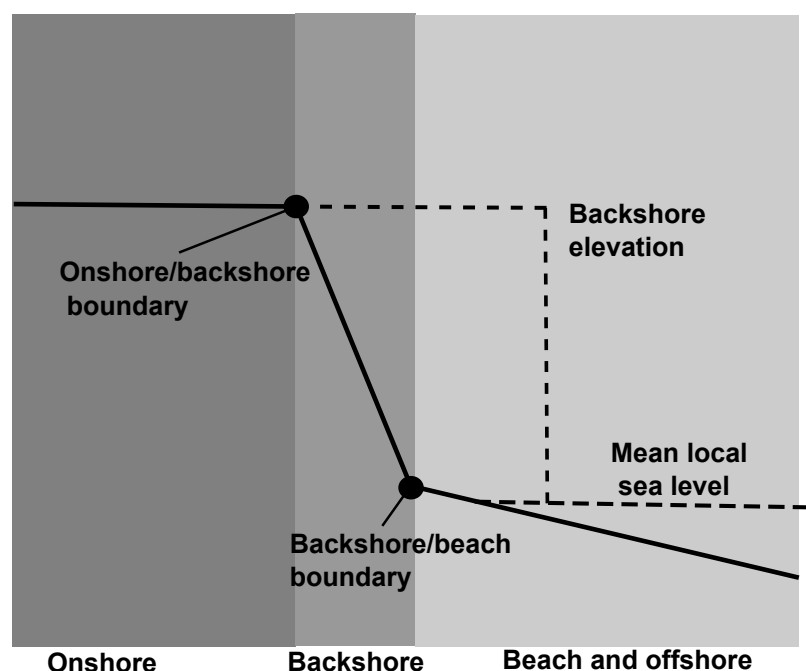

**Figure A3.** Illustration of the boundaries and elevation estimates described in Section 2.4

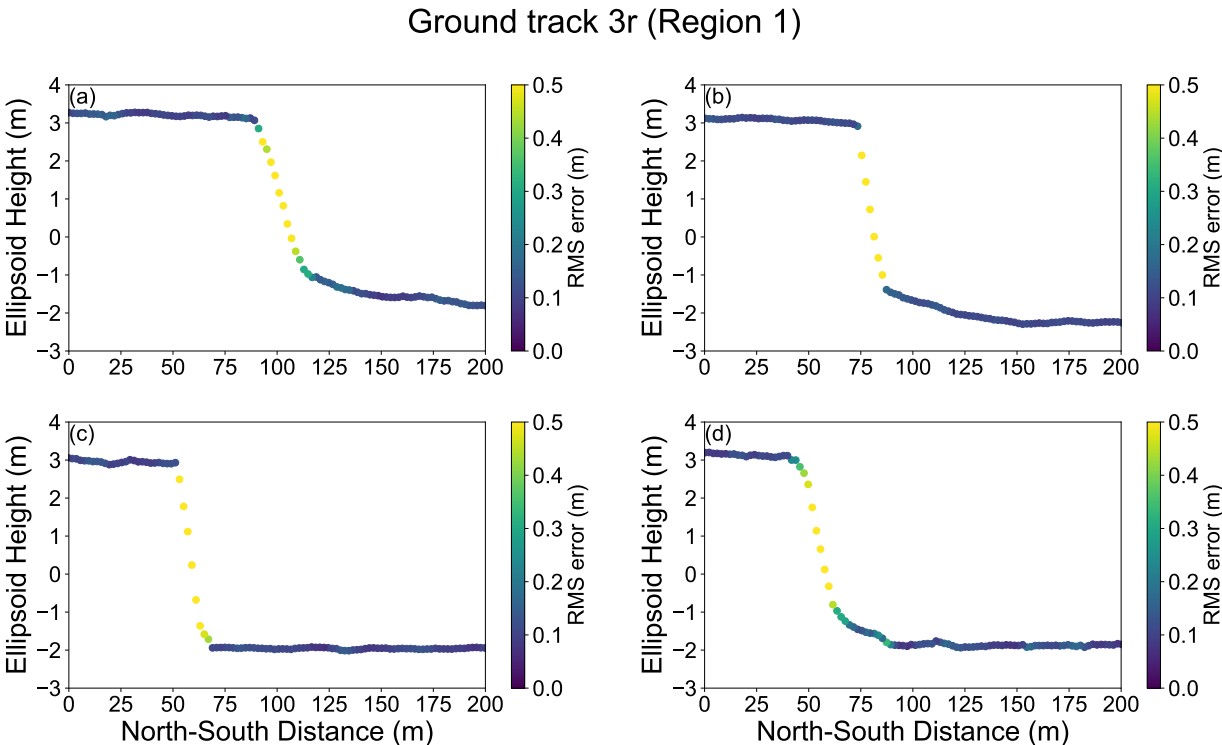

**Figure A4.** RMS error between our SlideRule-derived elevations and the source ATL03 photon information for ground track 3r for (a) 7 April 2019; (b) 4 January 2020; (c) 2 July 2021; (d) 31 December 2021





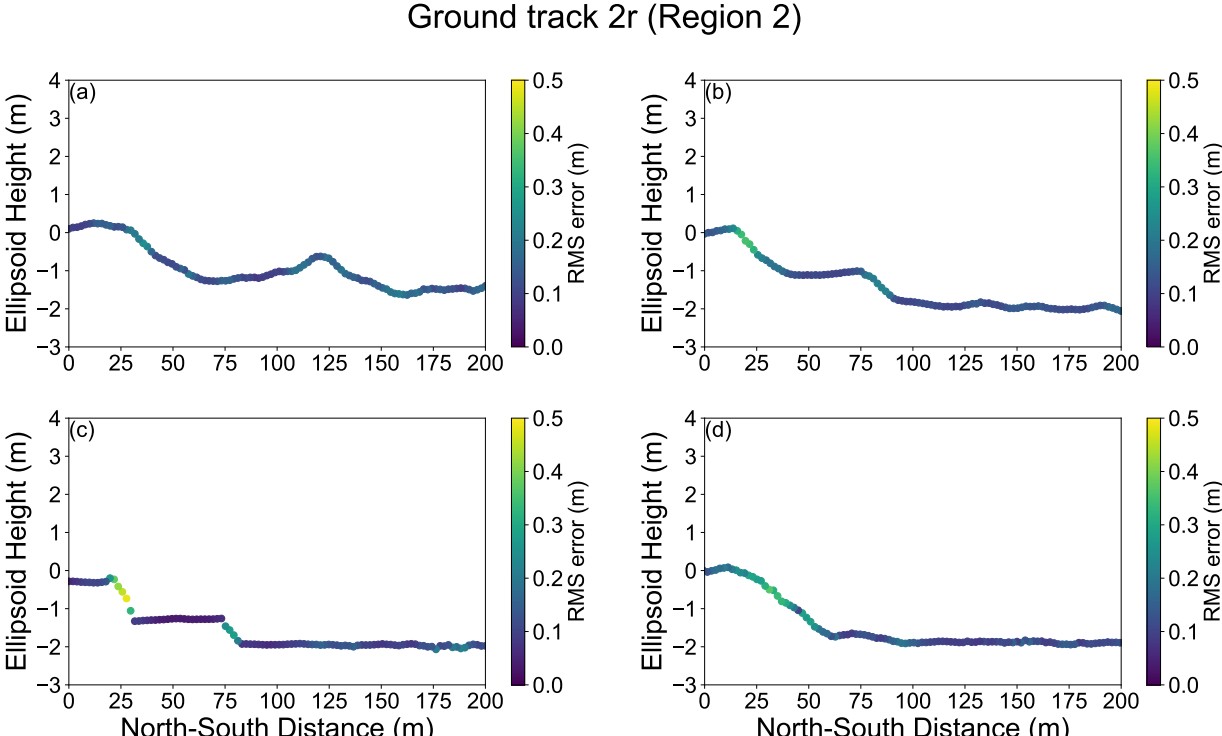

**Figure A5.** RMS error between our SlideRule-derived elevations and the source ATL03 photon information for ground track 2r for (a) 7 April 2019; (b) 4 January 2020; (c) 2 July 2021; (d) 31 December 2021




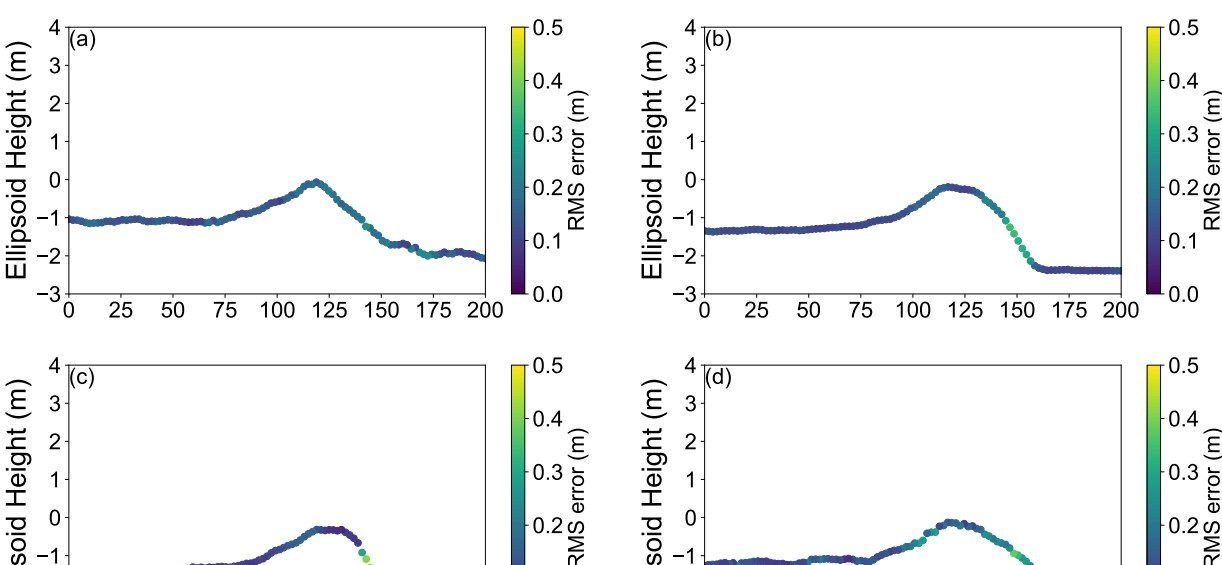

**Figure A6.** RMS error between our SlideRule-derived elevations and the source ATL03 photon information for ground track 1r for (a) 7 April 2019; (b) 4 January 2020; (c) 2 July 2021; (d) 31 December 2021



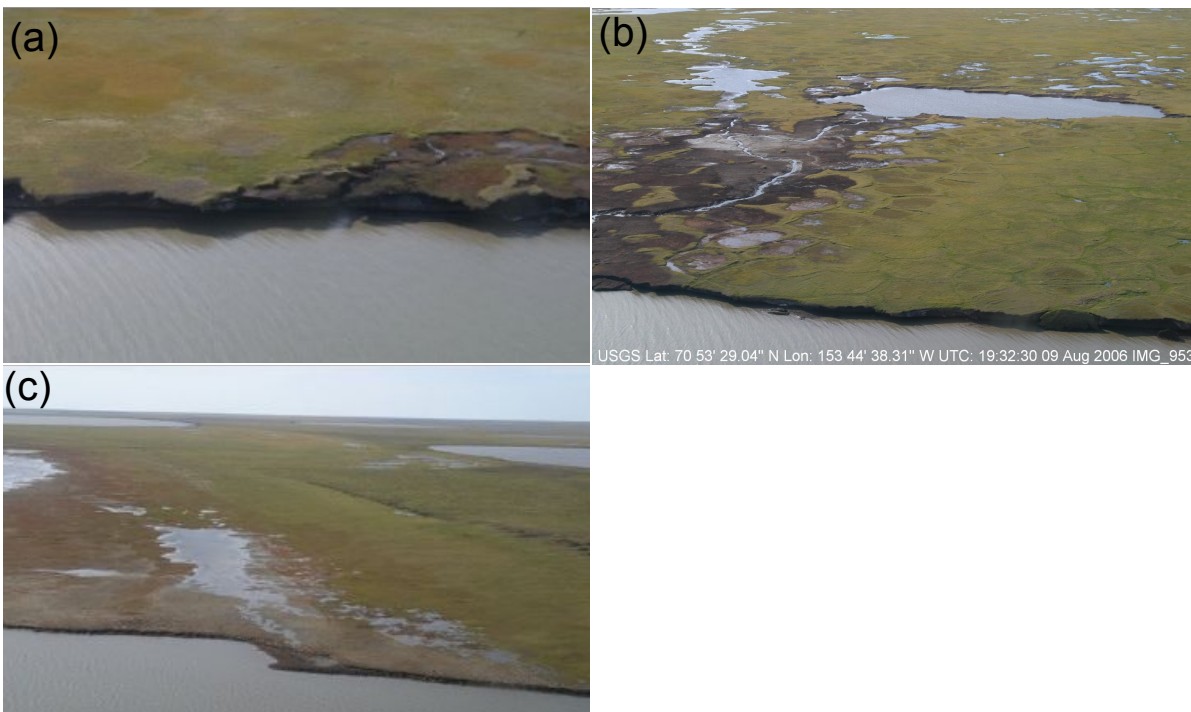

**Figure A7.** 2007 aerial photography from NOAA ShoreZone (Harper and Morris, 2014) captured near the sites our ICESat-2 ground tracks. a) A coastal bluff near the sampling site of ground track 3r in Region 1. b) The small lake sampled by ground track 2r in Region 2 before it drained. c) The western edge of the large breached thermokarst lake in Region 3 that is crossed by ground track 1r.



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
