# Peer review of "Multiple modes of shoreline change along the Alaskan Beaufort Sea observed using ICESat-2 altimetry and satellite imagery"

_EGUsphere, 2024_

## Referee Comment (RC1)

**Review of**
**Multiple modes of shoreline change along the Alaskan Beaufort Sea**
**observed using ICESat-2 altimetry and satellite imagery**
**Bryant et al. 2024-1656**

This article presents estimation of the retreat rate of an 8 km shoreline in Alaska from time series of satellite optical images (Planet) and high-resolution satellite laser altimetry (ICESat-2). The retreat is estimated for 3 years continuously along the coastline (imagery) and along three transects (altimetry). Both methods results in similar rates estimate and highlights the interannual and spatial variability of the retreat patterns. The processes potentially leading to these retreat patterns are explored.

I appreciated reading this article as it is well written, presented and concise. The methods are well explained and make good use of novel datasets. I have no background on the specific topic of coastal dynamic and cannot evaluate the quality or novelty of this work to this regard. However, from the introduction, it sounds like this is the first work using ICESat-2 data at such spatial resolution to estimate coastal retreat rate. If this is the case, it should be emphasized as a novelty of this article. Furthermore, I suggest the authors to consider the following improvement before considering the article ready to be published.

César Deschamps-Berger

**L8** Speed formating See the Cryosphere Author Guideline : "(e.g. 10 km h$^{-1}$ instead of 10 km/h)."
**L12** : "*Our topographic profiles from ICESat-2 highlight three distinct shoreline types...*" Are the shoreline types really distinguished from the ICESat-2 data? It seems more like an optical images analysis. Maybe as well move this sentence before the previous sentence.
**L15** "*can provide*" => "*provide*" (if it did, of course)
**L20** Hard to read, citations should be moved at the end of the sentence.
**L29** "*During the open water season,  i.e. the coasts are not sheltered by sea ice*"
**L34** "*to be highly variable on local scales (∼10s of meters)*" at what temporal scale are the rate variable ? Decadal like for the regional scale rates ? Or on shorter term ? I think it is important to always specify the spatial and temporal scale of the changes considered.
**L67** "*Satellite-based…*" unclear if this is what will be developed in this article or pre-existing studies? In the latter case, cite studies. For instance, is there no work based on the ArcticDEM dataset?
**L76** "*cm-to-dm*" write in full letter, dm is not so clear
**L79** "*sub-satellite ground track*" ?
**L80** "*repeat-track mode*" what other mode is there?
**L83** "*water(Jasinski*" missing space
**L85** "*< 10 m*" write with words
**L88** "*photon data (ATL03)*"
**L93** "*Jones et al. (2009)*" I would delete this to alleviate the ()
**L110** "*(with negative shoreline change indicating retreat)*" to move at the first occurrence of rates description in the text
**L112** "*storm occurrence,  storm power*" ?

**L117** "*by CNES Airbus*" maybe give the satellite name. From a rapid check on
https://www.intelligence-airbusds.com/en/4871-ordering, I get the feeling that what is shown on
Google Earth might be a mosaic of a Pléiades and SPOT-6-7 images possibly on 19-09-2018
https://www.intelligence-airbusds.com/satellite-image/?
id=DS_PHR1A_201809192213155_FR1_PX_W154N70_0219_03392
https://www.intelligence-airbusds.com/satellite-image/?
id=DS_SPOT7_201809192147155_FR1_FR1_FR1_FR1_W153N71_02602

**L120** "*composition*" ? Could it be more precise? geometry ?

**L124** "*by Gibbs and Richmond (2015) and Jones et al. (2009)*" a bit too much importance given to
citations, makes reading complicated in this part.

**L130** "*as an inundated*"?

**L130** "*by low elevations*" isn't this characteristic of all the area ? Maybe give the range of elevation
in the area in Study Site.

**L156** "*when retreat when ocean*" ?

**L162** "*time interval*" ? Interval between two successive data acquisition?

**L172** "*using implemented in matplotlib contour in Python*" ?

**L197** "*(ground tracks 3r,2r, and 1r, labeled in Fig. 1 (a))*" this is, the most surprising
methodological point to me. Due to the switch between forward and backward orientation, the right
and left beam can be the strong or the weak beam. I understand that the time series obtained here is
composed of weak and strong beam data. Although I would not expect big differences between the
elevation of either beam, it should be at least commented and explained. As well, why only use the
right beam? Adding the left one would increase the data sampling and if too redundant, it would
provide an estimation of the uncertainty of the method.

**L207** "*the SlideRule Python Client*" I know that SlideRule is public but is the code of this article
available somewhere?

**L213** "*uncertainties propagated from ATL03*" what error field from ATL03 are used for this
uncertainty calculation?

**L215** Why using 80 % overlapp ? Sounds like a lot of repetitive data? Were other values tried (no
need to  reprocess anything if not)?

**L231** "*We identified the intersection between each ICESat-2 track and the corresponding imagery-
derived shoreline and compared the shoreline positions and north-south retreat estimates derived
from Planet and the two ICESat-2 boundaries*" I have one doubt: were the retreat from Planet and
ICESat-2 calculated along the same direction (the only one possible being the ICESat-2 track) for
the comparison?

**L241** "*find **that** they range*"?

**L245** "*2019*" This is just style and nothing mandatory but I would avoid starting a sentence with a
year. For instance, a few sentence further: "*in late October. 2021 saw*" is not easy to read. The "."
seems an error.

**L251** "*Imagery-derived shorelines **position and** retreat rates*" alternative title to avoid shoreline
repetition

**L258** "*corresponding to a **position** change  uncertainty of 3.1 m*"

**L258** : "*2.2 m, corresponding to a change estimate uncertainty of 3.1 m*" this assumes uncorrelated
error of both shoreline, maybe worth mentioning

**L260** "*Only 6 segments across the 3-year*" maybe give somewhere the total number of segments

**L264** "*Region 1 showed moderately high retreat*" Moderate or high? Sounds opposite.

**L267** "*with 15% of valid shoreline segments*" maybe provide this metric for other years. It is hard to evaluate its meaning otherwise.

**L275** "*(-70.1 m of shoreline change)*" this made me think: could tides and waves have an impact on the shore detection (depending on the tide, wave amplitude and the bathymetry)?

**L288** "*we note that it consistently falls between the upper and lower*" is this result consistent with the errors estimated for both estimates (ICESat-2 and Planet). Do the error bars overlap?

**L306** "*that may correspond to toppled bluff material*" anything visible on the Planet imagery to back this hypothesis?

**L313** "*(Fig. A7(b))*" => "*(Fig. A7.b)*" I would avoid nested brackets.

**L314** "*in Airbus imagery from Google Earth (Fig. 6(c))*" for another study: could Landsat images be useful?

**L319** "a slight lowering by 0.23 m" I would avoid brackets as much as possible to ease the reading.

**L334** "*are higher than long-term historical estimates and similar to recent observations*" a visual way to represent that (for future work or here it seems relevant) could be to represent with lines or rectangles previous results and results of this article on the same timeline (x axis being time, y the position change). As is done for glacier mass balance. For instance, see Fig. 8 in Falashi et al. 2023 (https://tc.copernicus.org/articles/17/5435/2023/). Rectangles instead of lines allow to show range or uncertainty.

**L416** "*in **the** 90ᵗʰ*"?

**L473** "*AK*" => "*Alaska*"

**L474** "*We found  km-scale variability in shoreline **annual** change*"?

**Figure 4** If I guess correctly: add in the caption that the dashed lined are drawn assuming stable shore position during ice-on periods and evolving linearly during ice-free period?

**Figure 5** Add the 1:1 line.

**Figure 6** It could be useful to show a Planet image as background on the left pannel. It is a bit confusing to see the shoreline more advanced into the sea, even more with the 2024 copyright date. Maybe as well zooming in a bit more? It is hard to get information from the background image at this resolution.

---

## Referee Comment (RC2)

**Review of "Multiple modes of shoreline change along the Alaskan Beaufort Sea observed using ICESat-2 altimetry and satellite imagery" by Bryant et al. 2024**

**Summary**
This work makes a significant contribution to Arctic coastal erosion research, and particularly the use of ICESat-2 for coastal applications. The authors use annual PlanetScope imagery-derived shorelines (NDWI/Otsu) and ICESat-2 backshore and shoreline (manual) locations. Open water days, cumulative wave energy, and other environmental variables are brought in from ERA5 to better understand drivers of erosion. Fine-scale features are visible in the ICESat-2 photon data. A 10.7m/a shoreline erosion rate for Drew Point between 2019 and 2021 was reported, but also contextualized within recent decades of work, the outlier of the 2019 season, and local variability in shore classification. Slope measurements from ICESat-2 are discussed in the context of erosion rates/classifications from other sources.

The key contribution is the novel application of fine-grained photon-level analysis for coastal settings, especially as a complement to optical satellite imagery-based estimates of shoreline change. Importantly, the authors provide a thorough discussion of ICESat-2 uncertainty and leverage the repeating ground tracks in the Arctic for unique measurements of change from elevation profiles, comparing them to imagery-derived estimates. While Drew Point is an outlier for its high change rates, these same rates make it particularly valuable for honing satellite-based Arctic coastal change methods, and this work makes a notable contribution by focusing primarily on satellite data. Features observed in the ICESat-2 data are thoroughly explained and used to explain/compared to erosion rates. In terms of the applicability of ICESat-2 for shoreline monitoring, the upper shoreline is shown to better match the Planet-derived shoreline estimates.

Section/Paragraph Level Response
- 2.3 Overall, this section could benefit from at least some citing of the existing, and especially recent research into sub-pixel shoreline extraction from satellite imagery. I think this method is sound and the thresholds in the Appendix are acceptable, but there's enough variation in the literature I'm curious why you went with what you did. Perhaps existing tools like CoastSat do best with sandy beaches with no sea ice, and a simpler approach does fine here. Or perhaps existing tools were challenging to integrate with PlanetScope? Would this work for other locations along the Beaufort Sea Coast? In any case I think that's worth clarifying to future readers, even if this paper is focusing more on ICESat-2 than rehashing satellite shoreline methods, which is understandable.
- 2.3. I'm convinced by your argument of the North-South simplifying assumption for this study site. However, this is likely only generally applicable for Arctic coasts, and even then, I'm not sure if the associated uncertainty of this assumption would be a problem for anywhere rates of erosion are much lower than Drew Point. Maybe a sentence here or in the discussion better clarifying why you opted not to go with standard cross-shore transects, or whether this is a valid assumption for Beaufort coast locations other than Drew Point.
- 2.3.P4 More explanation is needed about how and why you used matplotlib contour.
- 2.4.P2 (/Introduction) I agree the terrain heights provided by ATL08 are too coarse, but there are ground/vegetation classification data provided at photon resolution, and easily filterable using SlideRule. It's possible that these classifications are over-smoothing coastal features here and shouldn't be used but could be worth showing/saying so if that's the case.

Similarly, why or why not use quality_ph flags that come with ATL03 to filter afterpulsing, instead of manually applying a 0.8m cutoff?

- 2.4.P2/P3. I am curious about whether signal_conf_ph > 3 is sufficiently including photons from the face of the bluff. I doubt this warrants anything like a new plot, but given the manual inspection/selection of the shoreline features, it makes sense to mention why that threshold was selected, even if cursorily. The custom ATL06 processing you use should help filter out most, if any errors a lower confidence threshold might introduce, while including more photons, potentially improving the accuracy of slope measurements.
- 2.4.P4 It's not clear when the custom-ATL06 derived heights were used or when the photon data was used. It seems like the photon data was only used to generate the custom-ATL06 heights, and for the discussion of visible features, while the ATL06 derived-heights were used for the upper and lower shoreline detection. Maybe a small clarification here would help. Similarly, if you suspected the photon data was a toppled bluff, how should you address the manual classification of the beach/backshore?
- 2.4 Did you consider trying to classify the instantaneous waterline from the ICESat-2 profiles? Perhaps this was beyond the scope of the study, but it's not clear whether the waterline might be detectable from the ICESat-2 photon data. If only because the Planet shorelines are instantaneous waterlines, but the ICESat-2 shorelines technically aren't, I think it may be worth addressing.
- 2.4 Please clarify if you are using strong beams, weak beams, or both.
- Figure 5. A 1:1 line would be helpful for comparison.
- 4.2.P1 It's great to see the incorporation of environmental variables along-side ICESat-2 and Planet-based erosion estimates, and the annual trends are clear. Are these valuable at a finer-scale, either temporally or spatially, for similar or other study areas? I was expecting there would be more analysis of these data compared to the shoreline change rates. For example, are storm events and their corresponding increases in erosion rates measurable within a season from some combination of Planet/ICESat-2? Any numbers regarding the temporal variability of erosion rates with respect to environmental variables might complement the existing paragraph well.

**Stylistic comments**
The manuscript is well organized, edited, and the points to be made were clear. Figures are clear and well-designed. The following minor typos were found.
- L72. Missing space after citation.
- L83. Missing space before citation.
- L155. Needs rephrasing.
- L368. Shoreline misspelled.
- L404. Missing space before parenthesis.

---

## Author Comment (AC1)

(Note: author responses are in italics)

 **L215** Why using 80 % overlap ? Sounds like a lot of repetitive data? Were other values tried (no need to reprocess anything if not)?

*High along-track spacing was desired in order to capture abrupt elevation changes, such as the cliff face in track gt3r. However, a short segment length provides fewer photons for the linear fit and is more sensitive to along-track variations in photon density. A longer segment length results in a smoother profile, at the cost of capturing small-scale features. A 10-m segment length 2-m posting was chosen to strike a balance between these two considerations to give a smoothed profile with dense along-track sampling. We note that this 80% overlap does mean that consecutive along-track elevation segments are not independent of each other. The specific values were chosen due to providing a good visual fit with the underlying photon data. We acknowledge that future work would benefit from experimenting with different segment lengths and postings.*

 **L231** *"We identified the intersection between each ICESat-2 track and the corresponding imagery- derived shoreline and compared the shoreline positions and north-south retreat estimates derived from Planet and the two ICESat-2 boundaries"* I have one doubt: were the retreat from Planet and ICESat-2 calculated along the same direction (the only one possible being the ICESat-2 track) for the comparison?

*Rather than report the along-track retreat from ICESat-2, we calculated only the change in the north-south (vertical) component of the ICESat-2-derived shoreline position.*

 **L275** *"(-70.1 m of shoreline change)"* this made me think: could tides and waves have an impact on the shore detection (depending on the tide, wave amplitude and the bathymetry)?

*We thank the reviewer for bringing this to our attention. We note that the tides in this region tend to be less than 0.2 m, while storm surges can result in temporary relative sea level increases of up to 1.4 m (Jones et al, 2018). However, given that most of this region consists of steep bluffs with narrow, or no beaches (Gibbs and Richmond, 2015), changes in the local relative sea level are not expected to have a large impact on the observed land-water boundary. We expect that some variations in the shoreline due to changes in the instantaneous water level are captured in our uncertainty analysis (L179-L187). We will update the text to include this explanation.*

**L306** "*that may correspond to toppled bluff material*" anything visible on the Planet imagery to back this hypothesis?

*Planet imagery shows very little retreat at the site of the ICESat-2 location (-3.7 m c) as well as across the surrounding region (-7 m (± 3.1 m) across Region 1), which would be consistent with the presence of collapsed bluff material. However, due to the resolution and image quality of the Planet imagery, we are unable to pick out small features such as toppled blocks, such that we can't directly confirm this hypothesis with Planet imagery.*

**Figure 6** It could be useful to show a Planet image as background on the left panel. It is a bit confusing to see the shoreline more advanced into the sea, even more with the 2024 copyright date. Maybe as well zooming in a bit more? It is hard to get information from the background image at this resolution.

*We note that at the current spatial scale in the left hand side of figure 6, Planet appears over-zoomed, making it hard to distinguish small-scale shoreline features. Included below are examples of a Planet Image (from 7/25/2020, shown in figure 3a) plotted at the current zoom window of figures 6a, 6b, and 6c. Imagery from Google Earth was used as a higher-resolution alternative to provide a more detailed view. The 2024 copyright date is printed on the image to comply with Google Earth's attribution guidelines, but we acknowledge this is a bit confusing. We will add an additional annotation to each subfigure to denote that the source imagery was taken in 2018. The zoom level was set to be consistent across all 3 plots, and needed to be wide enough to include the entire drained lake basin in Figure 6c.*

[Figure]

[Figure]

[Figure]

**References**

Gibbs, A. and Richmond, B.: National assessment of shoreline change—Historical shoreline change along the north coast of Alaska, U.S.–Canadian border to Icy Cape, Open-File Report, U.S. Geological Survey, http://dx.doi.org/10.3133/ofr20151048, 2015.

Jones, B. M., Farquharson, L. M., Baughman, C. A., Buzard, R. M., Arp, C. D., Grosse, G., Bull, D. L., Günther, F., Nitze, I., Urban, F., Kasper, J. L., Frederick, J. M., Thomas, M., Jones, C., Mota, A., Dallimore, S., Tweedie, C., Maio, C., Mann, D. H., Richmond, B., Gibbs, A., Xiao, M., Sachs, T., Iwahana, G., Kanevskiy, M., and Romanovsky, V. E.: A decade of remotely sensed observations highlight complex processes linked to coastal permafrost bluff erosion in the Arctic, Environmental Research Letters, 13, 115001, https://doi.org/10.1088/1748-9326/aae471, 2018.

---

## Author Response (AR1)

Review comments are in black
Authors' responses are in blue, with proposed **new text** in bold. All line numbers refer to line numbers in the original/unrevised manuscript.

**Referee #1: César Deschamps-Berger**

This article presents estimation of the retreat rate of an 8 km shoreline in Alaska from time series of satellite optical images (Planet) and high-resolution satellite laser altimetry (ICESat-2). The retreat is estimated for 3 years continuously along the coastline (imagery) and along three transects (altimetry). Both methods results in similar rates estimate and highlights the interannual and spatial variability of the retreat patterns. The processes potentially leading to these retreat patterns are explored.
I appreciated reading this article as it is well written, presented and concise. The methods are well explained and make good use of novel datasets. I have no background on the specific topic of coastal dynamic and cannot evaluate the quality or novelty of this work to this regard. However, from the introduction, it sounds like this is the first work using ICESat-2 data at such spatial resolution to estimate coastal retreat rate. If this is the case, it should be emphasized as a novelty of this article. Furthermore, I suggest the authors to consider the following improvement before considering the article ready to be published

We thank the referee for the very thorough comments. We agree that the novelty of our application of ICESat-2 should be emphasized, and propose the following changes to achieve this:

-**L3**: "Here, we use **a novel combination** of shoreline boundaries from multispectral imagery from Planet and topographic profiles from ICESat-2 satellite altimetry to compare year-to-year changes in shoreline position and morphology across different shoreline types "

-**L67-70**: Move the last sentence of the preceding paragraph to the beginning of the next paragraph:

" Elevation measurements from airborne lidar (e.g., Jones et al., 2013) and aerial photogrammetry (e.g., Gibbs et al., 2019; Lim et al., 2020a, b) can be used to qualitatively characterize the shoreline, provide high-resolution estimates of shoreline position, capture short-term topographic change, and enable comparisons of retreat rates between different geomorphic units (e.g., Lim et al., 2020a) on seasonal (e.g., Gibbs et al., 2019; Lim et al., 2020a) to multi-year (e.g., Jones et al., 2013) timescales and over km-scale areas e.g., Lim et al. 2020a). .

        The Ice, Cloud and land Elevation Satellite 2 (ICESat-2) laser altimeter collects repeat cross-shore elevation profiles, **providing the potential to expand on previous**

**elevation-based work with satellite altimetry and transform our understanding of Arctic shoreline morphology and change...**"

-**L88**: Add a new topic sentence to this paragraph: " **Here, we present a case study demonstrating how repeat altimetry from ICESat-2 can be used in tandem with satellite imagery to track annual shoreline change and provide insight on short-term and local shoreline processes.**"

-**L445**: Add a new topic sentence: "**Analyses of both the geolocated photon data and derived elevation profiles from ICESat-2 provide valuable insight on shoreline change.**"

**L8** Speed formating See the Cryosphere Author Guideline : "(e.g. 10 km h-1 instead of 10 km/h)."

We thank the reviewer for bringing this to our attention and have updated the formatting for all rates mentioned in the manuscript.

**L12** : "*Our topographic profiles from ICESat-2 highlight three distinct shoreline types..*" Are the shoreline types really distinguished from the ICESat-2 data? It seems more like an optical images analysis. Maybe as well move this sentence before the previous sentence.

We realize the wording here is ambiguous, and have rephrase it as:

"Our topographic profiles from ICESat-2 **sample** three distinct shoreline types "

**L15** "*can provide*" => "*provide*" (if it did, of course)

We have changed 'can provide' to 'provide'

**L20** Hard to read, citations should be moved at the end of the sentence.

Although we realize this impacts readability, the citations are currently placed to make it clear which references are associated with each stated environmental process.

**L29** "*During the open water season, when **i.e.** the coasts are not sheltered by sea ice*"

We have rephrased this sentence as: "During the open water season, **i.e.,** when the coasts are not sheltered by sea ice."

**L34** "*to be highly variable on local scales (~10s of meters)*" at what temporal scale are the rate variable ? Decadal like for the regional scale rates ? Or on shorter term ? I think it is important to always specify the spatial and temporal scale of the changes considered.

We thank the reviewer for pointing this out and have updated this sentence to clarify the timescales:

" studies that consider the spatial distribution of **decadal and annual** retreat rates have found them to be highly variable on local scales (~10s of meters) (Gibbs and Richmond, 2015; Farquharson et al., 2018; Irrgang et al., 2018; Jones et al., 2018)."

**L67** "*Satellite-based*..." unclear if this is what will be developed in this article or pre-existing studies? In the latter case, cite studies. For instance, is there no work based on the ArcticDEM dataset?

We are not aware of any work using Arctic DEM for coastal studies. We have cited a couple of previous ICESat-2-based studies in the next paragraph (L77-78). We have also made the following change to make the transition between previous elevation-based work and our planned work more clear:

" Elevation measurements from airborne lidar (e.g., Jones et al., 2013) and aerial photogrammetry (e.g., Gibbs et al., 2019; Lim et al., 2020a, b) can be used to qualitatively characterize the shoreline, provide high-resolution estimates of shoreline position, capture short-term topographic change, and enable comparisons of retreat rates between different geomorphic units (e.g., Lim et al., 2020a) on seasonal (e.g., Gibbs et al., 2019; Lim et al., 2020a) to multi-year (e.g., Jones et al., 2013) timescales and over km-scale areas e.g., Lim et al. 2020a). .

      The Ice, Cloud and land Elevation Satellite 2 (ICESat-2) laser altimeter collects repeat cross-shore elevation profiles, **providing the potential to expand on previous elevation-based work and transform our understanding of Arctic shoreline morphology and change...**"

 **L76** "*cm-to-dm*" write in full letter, dm is not so clear

We have rewritten this to "centimeter-to-decimeter"

 **L79** "*sub-satellite ground track*" ?

We have simplified this to "ground track"

**L80** "*repeat-track mode*" what other mode is there?

We have removed the mention of "repeat-track mode" here and instead provided a brief explanation of ICESat-2 pointing modes in section 2.4:

**"Outside of the poles, the majority of ICESat-2 revisits between 2019 and 2022 were off-pointed from their nominal ground track location to increase areal coverage, such that subsequent revisits did not cover the same ground location. However, "Target of Opportunity" requests over the North Slope of Alaska**

**resulted in exact repeats of every fifth reference ground track starting in April 2019"**

**L83** "*water(Jasinski*" missing space

We have added a space

**L85** "*< 10 m*" write with words

We have rephrased this to be "sub-10 meter"

**L88** "*photon data (ATL03)*"

We have added "(ATL03)"

**L93** "*Jones et al. (2009)*" I would delete this to alleviate the ()

We have moved this citation to the end of the sentence for readability.

**L110** "*(with negative shoreline change indicating retreat)*" to move at the first occurrence of rates description in the text

We have moved this phrase up to the second sentence of the introduction, where retreat rates are first mentioned.

**L112** "*storm occurrence, and storm power*" ?

We have rephrased this as "storm activity"

**L117** "*by CNES Airbus*" maybe give the satellite name. From a rapid check on https://www.intelligence-airbusds.com/en/4871-ordering, I get the feeling that what is shown on Google Earth might be a mosaic of a Pléiades and SPOT-6-7 images possibly on 19-09-2018 https://www.intelligence-airbusds.com/satellite-image/?id=DS_PHR1A_201809192213155_FR1_PX_W154N70_0219_03392 https://www.intelligence-airbusds.com/satellite-image/?id=DS_SPOT7_201809192147155_FR1_FR1_FR1_FR1_W153N71_02602

We agree that the images listed above may be the source imagery for the Google Earth mosaic image we used. However, we have been unable to definitively confirm whether our image is derived from Pleiades, SPOT, or both, and as such feel that it is appropriate to just use the general CNES Airbus attribution provided by Google Earth.

**L120** "*composition*" ? Could it be more precise? geometry ?
We have rephrased this as:

"These classifications are based on **the substrate and morphology** of the shoreline as well as dominant erosion and accretion processes thought to be present."

**L124** "*by Gibbs and Richmond (2015) and Jones et al. (2009)*" a bit too much importance given to citations, makes reading complicated in this part.
We agree this is a bit confusing, and have rephrased this as:

"Region 1, the westernmost portion of the study area, primarily consists of steep, ice-rich coastal bluffs (Jones et. al, 2009, Harper and Morris, 2014, Gibbs and Richmond, 2015)."

**L130** "*as an inundated*"?
We have corrected this typo

**L130** "*by low elevations*" isn't this characteristic of all the area ? Maybe give the range of elevation in the area in Study Site.
We acknowledge that this is a bit vague, and have clarified this statement:

"The easternmost portion of this shoreline is classified as an inundated tundra environment, **where the nearshore elevation is below sea-level, and there has been significant thaw subsidence and flooding**."

We have also added the typical shoreline elevations in this region after L105:

"**Most of the shoreline in this region is less than 3 m high, although the bluffs near drew point reach as high as 6 m (Gibbs and Richmond, 2015)**"

**L156** "*when retreat when ocean*" ?
We have removed "when retreat", which was a typo

**L162** "*time interval*" ? Interval between two successive data acquisition?
We have clarified this sentence:

"We also recorded the mean air and ocean temperature **between 1 June and 31 October of each year**"

**L172** "*using implemented in matplotlib contour in Python*" ?
This was a typo, and we've added in the missing text:
"We identified the sub-pixel land-water boundary from our NDWI images using **a marching squares algorithm** implemented in matplotlib contour in Python"

**L197** "*(ground tracks 3r,2r, and 1r, labeled in Fig. 1 (a))*" this is, the most surprising methodological point to me. Due to the switch between forward and backward orientation, the right and left beam can be the strong or the weak beam. I understand that the time series obtained here is composed of weak and strong beam data. Although I would not expect big differences between the elevation of either beam, it should be at least commented and explained. As well, why only use the right beam? Adding the left

one would increase the data sampling and if too redundant, it would provide an estimation of the uncertainty of the method.

We thank the reviewer for pointing this out, and we acknowledge that the time series is a combination of strong and weak beam data. Brunt et. al (2019) found the elevation difference between strong/weak beams in a pair to range from 0.6 cm (ground tracks 2l and 2r) to 1.3 cm (ground tracks 3l and 3r). These elevation biases, which are based on analysis of ICESat-2 release 001 data so are likely maximum estimates, are small compared to other potential sources of elevation change (such as changes in snowcover and morphologic change), and we do not anticipate that they would impact our estimates of horizontal shoreline position. We have updated Table 1 to show which acquisitions correspond with strong vs weak beams (see below).

| Interval | Planet dates | ICESat-2 dates |
|----------|--------------|----------------|
| 2019–2020 | 25 June 2019 – 25 July 2020 (154 owd) | 07 April 2019 (weak) – 04 January 2020 (strong) (138 owd) |
| 2020–2021 | 25 July 2020 – 02 July 2021 (97 owd) | 04 Janaury 2020 (strong) – 02 July 2021 (strong) (111 owd) |
| 2021–2022 | 02 July 2021 – 01 July 2022 (91 owd) | 02 July 2021 (strong) – 31 December 2021 (weak) (91 owd) |

When inspecting the available ICESat-2 data in this region, we found that for a given acquisition date, we sometimes only have "good" data (i.e. enough signal photons to derive a reliable surface) from either the left or the right beam. For example, the photon returns from the left beam on 2021-07-02 are insufficient to characterize the shoreline. Focusing our study on the right beam allowed us to ensure that we consistently had data at the same location on each sampling date, allowing for a more straightforward comparison of shoreline evolution between beams. Furthermore, focusing on just three beams allows for a more in-depth discussion of each individual beam. Future work will focus on bulk analysis of ICESat-2, such that all beam pairs for a given RGT are considered.

**L207** "*the SlideRule Python Client*" I know that SlideRule is public but is the code of this article available somewhere?
The full code is not available publicly at this time. We have tried to be explicit in describing our methods such that it can be replicated even without the exact code.

**L213** "*uncertainties propagated from ATL03*" what error field from ATL03 are used for this uncertainty calculation?
We have updated this for clarity:

"SlideRule provides the RMS error between the photons used in the final fitting and the final linear fit, as well as the photon-level elevation error that is propagated through the linear fit. **The photon elevation error is assumed to be uniform for a given segment and is estimated as the maximum of the segment RMS error and the background-corrected standard deviation.**"

**L215** Why using 80 % overlap ? Sounds like a lot of repetitive data? Were other values tried (no need to reprocess anything if not)?
To address this, we have added the following explanation:

"**For coastal applications, a high along-track resolution is preferable to capture abrupt elevation changes at the shoreline. However, shorter segments may not provide enough photons for a robust linear fit (particularly for low-albedo surface, such as the snow-free tundra), and may result in height estimates that are subject to along-track variations in photon density. A longer segment length results in a smoother profile, at the cost of not capturing small-scale features. In order to strike a balance between these two considerations,** we implemented the SlideRule ATL06 algorithm for 10 m long segments spaced every 2 m along track"

We note that this 80% overlap does mean that consecutive along-track elevation segments are not independent of each other. The specific values were chosen due to providing a good visual fit with the underlying photon data. We acknowledge that future work would benefit from experimenting with different segment lengths and postings.

**L231** "*We identified the intersection between each ICESat-2 track and the corresponding imagery- derived shoreline and compared the shoreline positions and north-south retreat estimates derived from Planet and the two ICESat-2 boundaries*" I have one doubt: were the retreat from Planet and ICESat-2 calculated along the same direction (the only one possible being the ICESat-2 track) for the comparison?
Rather than report the along-track retreat from ICESat-2, we calculated only the change in the north-south component of the ICESat-2-derived shoreline position. We have revised this to instead project our profiles in the direction perpendicular to the local shoreline, based on the assumption that the observed change is predominantly due to cross-shore movement. We have added L231-233 accordingly:

"We identified the intersection between each ICESat-2 track and the corresponding imagery- derived shoreline and compared the shoreline positions and  **cross-shore** retreat estimates derived from Planet and the two ICESat-2 boundaries. **Under the assumption that the majority of the observed position change in our ICESat-2 boundaries is due to cross-shore change, we projected our ICESat-2-derived change estimate into the local cross-shore direction. The local cross-shore vector was based off of the baseline transect (as defined in Section 3.2) that was located closest to the geographic midpoint of the four ICESat-2 transects associated with each ground track.**"

**L241** "*find **that** they range*"?
We have added the word "that"

**L245** "*2019*" This is just style and nothing mandatory but I would avoid starting a sentence with a year. For instance, a few sentence further: "*in late October. 2021 saw*" is not easy to read. The "." seems an error.

We have rephrased this section as:

" **We observed the most extreme environmental conditions by all measured metrics in 2019** (Table 2). **The 2019 erosion year** had the longest open water season, with sea ice breaking up sooner (late June) and re-forming later (mid-November) than in the other 2 years"

The '.' was a typo and has been removed.

 **L251** "*Imagery-derived shorelines **position and** retreat rates*" alternative title to avoid shoreline repetition
 We agree that this is a clearer title and have updated it.

 **L258** "*corresponding to a **position** change estimate uncertainty of 3.1 m*"
We have updated add the word "position"

 **L258** : "*2.2 m, corresponding to a change estimate uncertainty of 3.1 m*" this assumes uncorrelated error of both shoreline, maybe worth mentioning
 We agree that this is worth mentioning and have revised this section to:

"Based on the uncertainty estimation described in Section 2.3, we estimated the precision of our shoreline positions estimates to be 2.2 m. **Assuming that the error in each shoreline is independent of the others,** this corresponds to a change estimate uncertainty of 3.1 m."

 **L260** "*Only 6 segments across the 3-year*" maybe give somewhere the total number of segments
In response to other reviewer comments, we have updated our analysis to calculate shoreline change in the cross-shore section. As a result, this section has been removed.

**L264** "*Region 1 showed moderately high retreat*" Moderate or high? Sounds opposite.
We've removed the word 'moderately'

**L267** "*with 15% of valid shoreline segments*" maybe provide this metric for other years. It is hard to evaluate its meaning otherwise.
We have updated the text to clarify:

" **While every shoreline segment in region 1 underwent retreat in 2019 and 2020,** 16% of valid shoreline segments (all in the western half of Region 1) did not exhibit substantial (> 3.1 m) shoreline change in 2021."

 **L275** "*(-70.1 m of shoreline change)*" this made me think: could tides and waves have an impact on the shore detection (depending on the tide, wave amplitude and the bathymetry)?
We thank the reviewer for bringing this to our attention. We have added following passage to Section 2.3:

**"The tides in this region tend to be less than 0.2 m, but storm surges can result in temporary relative sea level increases of 1.4 m (Jones et al, 2018). However, given that much of this region consists of steep bluffs with narrow, or no beaches (Gibbs and Richmond, 2015), changes in the local relative sea level are not expected to have a large impact on the observed land-water boundary."**

We expect that variations in the shoreline due to changes in the instantaneous water level are captured in our uncertainty analysis (L179-L187).

**L288** "*we note that it consistently falls between the upper and lower*" is this result consistent with the errors estimated for both estimates (ICESat-2 and Planet). Do the error bars overlap?
We agree that this is important to clarify. We have updated the text to give the upper and lower bounds of the difference between the Planet and ICESat-2 derived boundaries as well as their associated uncertainties:

"We would expect the land-water boundary from Planet to be located seaward (north) of the lower backshore boundaries identified by ICESat-2. However, we found that the Planet-derived land-water boundary consistently falls landwards of the ICESat-2-derived lower boundary **(by 8.6 ± 4.2 m to 41.1 ± 4.5 m), and either seawards (by up to to 36.0 m ± 4.5 m) or slightly landward of the ICESat-2-derived upper boundary (by up to 3.3 ± 4.0 m)."**

**L306** "*that may correspond to toppled bluff material*" anything visible on the Planet imagery to back this hypothesis?
Planet imagery shows little retreat at the site of the ICESat-2 location (-3.7 m) as well as across the surrounding region (-7 m (± 3.1 m) across region 1), which would be consistent with the presence of collapsed bluff material. However, due to the resolution and image quality of the Planet imagery, we are unable to identify small features such as toppled blocks, such that we can't directly confirm this hypothesis with Planet imagery.

**L313** "*(Fig. A7(b))*" => "*(Fig. A7.b)*" I would avoid nested brackets.

We have updated our in-text figure citations to avoid nested brackets

**L314** "*in Airbus imagery from Google Earth (Fig. 6(c))*" for another study: could Landsat images be useful?

Landsat images could be useful at looking at long-term retreat in this region, but the lower resolution would limit its use for short-term change and the more detailed shoreline analysis done here.

**L319** "a slight lowering (by 0.23 m)" I would avoid brackets as much as possible to ease the reading.

We have made the following changes to try to avoid over-using using parentheses in this section:

-**L310**: "The backshore elevation remains stable**,** ranging from 4.89 m to 5.02 m"

-**L311**: "...the remnant basin of a ~150 m  diameter lake that was breached"

-**L316**: " This resulted in a 0.40 m drop in the backshore height from 1.33 m to 0.93 m"

-**L319**: " We also observe a slight lowering  **of** 0.23 m of the backshore height"

-**L335**: "...passes over a 1.71 m high dune in front of a ~ 2.6 km  **km-wide** breached thermokarst lake"

-**L328**: "We note a 0.22 m drop in the backshore elevation from 1.89 m to 1.67 m between 2019 and 2020, after which the elevation remains stable from 2020 to late 2021 (**ranging from** 1.64 m to 1.67 m)"

**L334** "*are higher than long-term historical estimates and similar to recent observations*" a visual way to represent that (for future work or here it seems relevant) could be to represent with lines or rectangles previous results and results of this article on the same timeline (x axis being time, y the position change). As is done for glacier mass balance. For instance, see Fig. 8 in Falashi et al. 2023 (https://tc.copernicus.org/articles/17/5435/2023/). Rectangles instead of lines allow to show range or uncertainty.

We agree that a figure would help illustrate our rates compared to historical rates and have added this figure to the supplement: (**Figure A7** in the updated manuscript).

[Figure]

**Figure A7: Long and short-term regional shoreline change rates along the Alaskan Beaufort Sea Coast near Drew point from previous work, along with the 3-year and year-to-year regional rates derived in this work**.

We note that this figure appears to show increases in retreat rates over time, but that more recent studies (including ours) cover shorter time intervals than older studies. Since retreat is highly episodic, retreat rates estimated over long periods may be averaging periods of high and low retreat, leading to a lower overall rate compared to short-term estimates. While this figure provides a useful visual to put our results into the context of previous work, it also may lead readers to draw conclusions about trends in retreat rates over time that we are not trying to address in this work. Therefore, wel include this figure in the supplement, and have updated the main text (L339-342) to acknowledge the apparent trend while also acknowledging there are multiple potential explanations for it:

" **Our 3-year mean of observed retreat rates (-16.5 m) is higher than the long-term estimates of Gibbs and Richmond (2015) and Jones et al. (2009) and similar to the decadal-scale estimate from Jones et al. (2018).**  **While** our elevated estimates of erosion **compared to historical rates could be reflective of an increase in retreat rates over time**, they may also be due to the short time period of our observations, as short-term estimates of shoreline change tend to be  **higher in magnitude** than long-term estimates (Sadler and Jerolmack, 2015)."

**L416** "*in **the** 90th*"?

We have added the word "the"

**L473** "*AK*" => "*Alaska*"

We have changed "AK" to "Alaska"

**L474** "*We found annual km-scale variability in shoreline **annual** change*"?

We have changed this to "We found km-scale variability in **annual** shoreline change"

**Figure 4** If I guess correctly: add in the caption that the dashed lined are drawn assuming stable shore position during ice-on periods and evolving linearly during ice-free period?
This is correct, and we have added the following to our figure caption:
"Dashed lines indicate the trajectory of the shoreline based on a linear rate of change during the open water season and no change during ice-on periods"

**Figure 5** Add the 1:1 line.
We have added a 1:1 line to both figures (see below):

[Figure]

Figure 5. a) Comparison between shoreline change estimates from Planet and shoreline change estimates from the upper (orange) and lower
(blue) boundaries derived from ICESat-2. b) Comparison between the measured change between each open water season across the upper

and lower boundaries derived from ICESat-2. The coefficient of determination excluding the outlier drained lake measurement (grey) is
reported. Linear fits were estimated using orthogonal distance regression. **A 1-1 line is shown for reference**.

**Figure 6** It could be useful to show a Planet image as background on the left panel. It is a bit confusing to see the shoreline more advanced into the sea, even more with the 2024 copyright date. Maybe as well zooming in a bit more? It is hard to get information from the background image at this resolution.
We note that at the current spatial scale in the left hand side of Figure 6, Planet appears over-zoomed, making it hard to distinguish small-scale shoreline features. Imagery from Google Earth was used as a higher-resolution alternative. The zoom level was set to be consistent across all 3 plots, and needed to be wide enough to include the entire drained lake basin in Figure. The 2024 copyright date is listed to comply with Google Earth's attribution guidelines, but we have added an additional annotation to each subfigure to make it clearer that the source imagery was taken in 2018. An example is shown below:

[Figure]

**Referee #2: Anonymous**

**Summary**

This work makes a significant contribution to Arctic coastal erosion research, and particularly the use of ICESat-2 for coastal applications. The authors use annual PlanetScope imagery-derived shorelines (NDWI/Otsu) and ICESat-2 backshore and

shoreline (manual) locations. Open water days, cumulative wave energy, and other environmental variables are brought in from ERA5 to better understand drivers of erosion. Fine-scale features are visible in the ICESat-2 photon data. A 10.7m/a shoreline erosion rate for Drew Point between 2019 and 2021 was reported, but also contextualized within recent decades of work, the outlier of the 2019 season, and local variability in shore classification. Slope measurements from ICESat-2 are discussed in the context of erosion rates/classifications from other sources.

The key contribution is the novel application of fine-grained photon-level analysis for coastal settings, especially as a complement to optical satellite imagery-based estimates of shoreline change. Importantly, the authors provide a thorough discussion of ICESat-2 uncertainty and leverage the repeating ground tracks in the Arctic for unique measurements of change from elevation profiles, comparing them to imagery-derived estimates. While Drew Point is an outlier for its high change rates, these same rates make it particularly valuable for honing satellite-based Arctic coastal change methods, and this work makes a notable contribution by focusing primarily on satellite data. Features observed in the ICESat-2 data are thoroughly explained and used to explain/compared to erosion rates. In terms of the applicability of ICESat-2 for shoreline monitoring, the upper shoreline is shown to better match the Planet-derived shoreline estimates.

Section/Paragraph Level Response:

2.3 Overall, this section could benefit from at least some citing of the existing, and especially recent research into sub-pixel shoreline extraction from satellite imagery. I think this method is sound and the thresholds in the Appendix are acceptable, but there's enough variation in the literature I'm curious why you went with what you did. Perhaps existing tools like CoastSat do best with sandy beaches with no sea ice, and a simpler approach does fine here. Or perhaps existing tools were challenging to integrate with PlanetScope? Would this work for other locations along the Beaufort Sea Coast? In any case I think that's worth clarifying to future readers, even if this paper is focusing more on ICESat-2 than rehashing satellite shoreline methods, which is understandable. We note that the existing classes used in CoastSat (sand, whitewater, other) may not work properly with images that include sea ice and sediment-rich water as observed here. Although CoastSat may be a good alternative for shoreline detection in this region, we have not tested it. Our ndwi-based thresholding is more straightforward to implement than CoastSat and performed well enough on our test images that there was no need to use CoastSat as an alternative method. We note that while our method worked well on our four images, it may in general be susceptible to blunders due to clouds, offshore sediment, sea ice, and near-shore water. Additional testing would need to be done before we would recommend using it for other locations along the Beaufort Sea coast. We have added the following text:

-L168: "Historically, shoreline change has been estimated from satellites via manual delineation of the shoreline (Günther et al., 2015; Farquharson et al., 2018; Jones et al., 2009; Irrgang et al., 2018). Recent workflows such as CoastSat (Vos et al., 2019) have been developed to automatically detect the shorelines at

**sub-pixel resolution, but they have focused on lower-latitude beaches and may not perform well in Arctic regions where sea ice is present. Here, we implement our own shoreline detection method, following some of the same steps as Vos et al. (2019)."**

**-L173: " We found that calculating our threshold using all image pixels resulted in an adequate shoreline estimate for all four of our images, such that an initial identification of land and water pixels (as is done in Vos et al. (2019)) is not necessary."**

2.3. I'm convinced by your argument of the North-South simplifying assumption for this study site. However, this is likely only generally applicable for Arctic coasts, and even then, I'm not sure if the associated uncertainty of this assumption would be a problem for anywhere rates of erosion are much lower than Drew Point. Maybe a sentence here or in the discussion better clarifying why you opted not to go with standard cross-shore transects, or whether this is a valid assumption for Beaufort coast locations other than Drew Point.

Shoreline change was calculated in the north-south direction for ease of computation, and because the majority of the shoreline is approximately east-west oriented, such that the north-south is approximately equal to the cross-shore direction. However, based on feedback from reviewers, we have decided to update our shoreline change estimates to be calculated in the local cross shore direction. We have added the following text detailed the new shoreline change estimation method:

"Finally, we estimated shoreline change **using an approach similar to the USGS DigitalShoreline Analysis Software (Himmelstoss et al., 2021). In this workflow, a reference shoreline (hereafter referred to as the baseline) is selected. Cross-shore transects are generated at regular intervals along-shore that are perpendicular to this baseline. The change between consecutive shorelines is then calculated along these transects. We created a baseline by smoothing theraw 2020 shoreline with a 60 m along-shore running mean in order to assure adjacent cross-shore transects did not intersect onthe time interval of our study site. Transects were generated every 10 m along the baseline, and the change along each transect was calculated between each successive shoreline.**"

As a result of this change, we have removed L188-195 and Figure A2.

2.3.P4 More explanation is needed about how and why you used matplotlib contour.
We have updated this sentence to include what matplotlib contour does:

"We identified the sub-pixel land-water boundary from our NDWI images using **a marching squares algorithm** implemented in matplotlib contour in Python"

2.4.P2 (/Introduction) I agree the terrain heights provided by ATL08 are too coarse, but there are ground/vegetation classification data provided at photon resolution, and easily

filterable using SlideRule. It's possible that these classifications are over-smoothing coastal features here and shouldn't be used but could be worth showing/saying so if that's the case.

While we agree that the ATL08 ground/vegetation classification may be useful for identifying signal photons, we expect that filtering photons the ATL08 ground classification would produce very similar results to using the ATL03 confidence scores as we did here.

Similarly, why or why not use quality_ph flags that come with ATL03 to filter afterpulsing, instead of manually applying a 0.8m cutoff?

Arndt and Fricker (2024, in review) found that the 0.45 m afterpulse (which is what we observe in Figure 7), is not reliably removed when filtering with the quality_ph flag. Thus, choosing a 0.8 m window (corresponding to 0.4 m above and below the surface) ensures that we are not including photons associated with the 0.55 m afterpulse. We have added Arndt and Fricker (2024, in review) as a reference in section 2.4 when discussing afterpulses:

Arndt, P. S. and Fricker, H. A.: A Framework for Automated Supraglacial Lake Detection and Depth Retrieval in ICESat-2 Photon Data Across the Greenland and Antarctic Ice Sheets, EGUsphere [preprint], https://doi.org/10.5194/egusphere-2024-1156, 2024.

2.4.P2/P3. I am curious about whether signal_conf_ph > 3 is sufficiently including photons from the face of the bluff. I doubt this warrants anything like a new plot, but given the manual inspection/selection of the shoreline features, it makes sense to mention why that threshold was selected, even if cursorily. The custom ATL06 processing you use should help filter out most, if any errors a lower confidence threshold might introduce, while including more photons, potentially improving the accuracy of slope measurements.

While we agree that any analysis of coastal ATL03 data should consider lower-confidence photons, we found that there were very few photons with signal_conf_ph > 3 on the backshore, such that our threshold removes outlier photons without removing useful photons.

2.4.P4 It's not clear when the custom-ATL06 derived heights were used or when the photon data was used. It seems like the photon data was only used to generate the custom-ATL06 heights, and for the discussion of visible features, while the ATL06 derived-heights were used for the upper and lower shoreline detection. Maybe a small clarification here would help. Similarly, if you suspected the photon data was a toppled bluff, how should you address the manual classification of the beach/backshore?

We have added the following sentences to clarify how the photon and custom ATL06 data was used:

-**L206: " These photon data were plotted and inspected to provide a qualitative analysis of small-scale features."**

-**L222:** " To estimate shoreline change **from our custom ICESat-2 elevation profiles...**"

Future work would benefit from removing photons that are thought to belong to toppled blocks prior to identifying the beach/backshore boundary.

2.4 Did you consider trying to classify the instantaneous waterline from the ICESat-2 profiles? Perhaps this was beyond the scope of the study, but it's not clear whether the waterline might be detectable from the ICESat-2 photon data. If only because the Planet shorelines are instantaneous waterlines, but the ICESat-2 shorelines technically aren't, I think it may be worth addressing.
This is partially addressed in L 223-225:

"The presence of sea ice and snow in three of the ICESat-2 tracks prevents the accurate identification of a land-water boundary. Instead, we identified the boundaries of the backshore, defined here as the relatively steep region between the beach or ocean and the onshore region."

We have added the additional justification on L226:
"**Since beaches in this region are very narrow when present (Gibbs and Richmond, 2015), we expect the lower boundary from ICESat-2 to be similar to the land-water boundary**."

2.4 Please clarify if you are using strong beams, weak beams, or both.
We have updated Table 1 to indicate which acquisitions came from strong vs weak beams.

| Interval | Planet dates | ICESat-2 dates |
|---|---|---|
| 2019–2020 | 25 June 2019 – 25 July 2020 (154 owd) | 07 April 2019 (weak) – 04 January 2020 (strong) (138 owd) |
| 2020–2021 | 25 July 2020 – 02 July 2021 (97 owd) | 04 Janaury 2020 (strong) – 02 July 2021 (strong) (111 owd) |
| 2021–2022 | 02 July 2021 – 01 July 2022 (91 owd) | 02 July 2021 (strong) – 31 December 2021 (weak) (91 owd) |

Figure 5. A 1:1 line would be helpful for comparison.
We have added a 1:1 line (see below)

[Figure]

Figure 5. a) Comparison between shoreline change estimates from Planet and shoreline change estimates from the upper (orange) and lower
(blue) boundaries derived from ICESat-2. b) Comparison between the measured change between each open water season across the upper
and lower boundaries derived from ICESat-2. The coefficient of determination excluding the outlier drained lake measurement (grey) is
reported. Linear fits were estimated using orthogonal distance regression. **A 1-1 line is shown for reference**.

4.2.P1 It's great to see the incorporation of environmental variables along-side ICESat-2 and Planet-based erosion estimates, and the annual trends are clear. Are these valuable at a finer-scale, either temporally or spatially, for similar or other study areas? I was expecting there would be more analysis of these data compared to the shoreline change rates. For example, are storm events and their corresponding increases in erosion rates measurable within a season from some combination of Planet/ICESat-2? Any numbers regarding the temporal variability of erosion rates with respect to environmental variables might complement the existing paragraph well.
ERA-5 output is available at hourly time-intervals, but its low resolution (~30 km) prevents analysis of spatial variability in shoreline change rates. While it may be possible to examine sub-seasonal shoreline change with Planet imagery, cloud cover makes the amount of images per year highly variable. We chose to limit the temporal analysis of this study to annual intervals because that is the shortest interval that could be reliably measured by both Planet and ICESat-2.

**Stylistic comments**
The manuscript is well organized, edited, and the points to be made were clear. Figures are clear and well-designed. The following minor typos were found.

L72. Missing space after citation.
We have added a space.

L83. Missing space before citation.
We have added a space.

L155. Needs rephrasing.
We have removed "when retreat", which was a typo

L368. Shoreline misspelled.
We have corrected this typo

L404. Missing space before parenthesis.
We have added a space

**Referee #3: Anonymous**

This is a very well written paper that explores shoreline change for a region of the Alaskan Beaufort Sea Coast near Drew Point. They utilize ICESat2 data alongside high-resolution Planet imagery to quantify coastal change rates and morphology. Through this exploration, they examine the processes controlling coastal change and variability in process and form along the shoreline in three different regions.
I found this paper extremely easy to read and follow. I often found myself writing a comment to suggest some addition, only to find that it was addressed in the next paragraph I read. Thus, my review is quite short, as this paper is clear and worthy of prompt publication.

My only significant comment is that the authors would benefit from more clearly and directly explaining the novelty and scientific advance. I think that lies in two areas: first in the use of ICESat2 to explore coastal change and second in the authors' ability to interpret mechanisms and processes of coastal change from remotely sensed data, which is often quite difficult if not impossible. These are both stated in the paper, but the language can be beefed up a bit in the intro, discussion, and conclusions to really emphasize this.

We thank the reviewer for the thorough and positive review, in particular for their suggestions of important additional reference material. We agree that the novelty of our use of ICESat-2 and satellite imagery to explore coastal processes should be emphasized, and propose the following changes:

-**L3**: "Here, we use **a novel combination** of shoreline boundaries from multispectral imagery from Planet and topographic profiles from ICESat-2 satellite altimetry to compare year-to-year changes in shoreline position and morphology across different shoreline types "

-**L67-70**: Move the last sentence of the preceding paragraph to the beginning of the next paragraph:

" Elevation measurements from airborne lidar (e.g., Jones et al., 2013) and aerial photogrammetry (e.g., Gibbs et al., 2019; Lim et al., 2020a, b) can be used to qualitatively characterize the shoreline, provide high-resolution estimates of shoreline position, capture short-term topographic change, and enable comparisons of retreat rates between different geomorphic units (e.g., Lim et al., 2020a) on seasonal (e.g., Gibbs et al., 2019; Lim et al., 2020a) to multi-year (e.g., Jones et al., 2013) timescales and over km-scale areas e.g., Lim et al. 2020a).

  The Ice, Cloud and land Elevation Satellite 2 (ICESat-2) laser altimeter collects repeat cross-shore elevation profiles, **providing the potential to expand on previous**

**elevation-based work and transform our understanding of Arctic shoreline morphology and change..."**

-**L88**: Add a new topic sentence to this paragraph: " **Here, we present a case study demonstrating how repeat altimetry from ICESat-2 can be utilized in tandem with satellite imagery to track annual shoreline change and provide insight on short-term and local shoreline processes.**"

-**L445**: Add a new topic sentence: "**Analyses of both the geolocated photon data and derived elevation profiles from ICESat-2 provide valuable insight on shoreline change.**"

-**L491**:Add sentence: "**By integrating satellite altimetry and multispectral imagery, we can study mechanisms of coastal change that have previously been challenging to identify with satellite remote sensing.**"

Other, more minor comments:

L91-92: I don't think this rate is representative of the entire Beaufort Sea Coast of Alaska. Isn't the average reported rate there on the order of -1 m/yr? This is according to the Gibbs and Richmond 2017 data. Erosion is locally very fast at Drew Point, but this region is not representative of the broader Alaskan Beaufort Sea coast (Piliouras et al., 2023).

We agree that this is an important distinction to make, and have updated this sentence:

" We focus on the  **the coastline surrounding Drew Point, Alaska,** where shoreline change rates are both high (averaging -22 m/a over the last decade) and variable (-48.8 m/a to 0 m/a on ~ 10 m length scales) (Jones et al., 2018"

The rate of retreat of the larger Beaufort Sea Coast Region is given in L23.

What is the reasoning/justification for calculating shoreline change in the north-south orientation rather than perpendicular to the local shoreline? This should be included in the paper.

Shoreline change was calculated in the north-south direction for ease of computation, and because the majority of the shoreline is approximately east-west oriented, such that the north-south is approximately equal to the cross-shore direction. However, based on feedback from reviewers, we have decided to update our shoreline change estimates to be calculated in the north-south direction. We have added the following text detailed the new shoreline change estimation method:

"Finally, we estimated shoreline change **using an approach similar to the USGS Digital Shoreline Analysis Software (Himmelstoss et al., 2021). In this workflow, a reference shoreline (hereafter referred to as the baseline) is selected. Cross-shore transects are generated at regular intervals along-shore that are**

**perpendicular to this baseline. The change between consecutive shorelines is then calculated along these transects. We created a baseline by smoothing the raw 2020 shoreline with a 60 m along-shore running mean in order to assure adjacent cross-shore transects did not intersect on the time interval of our study site. Transects were generated every 10 m along the baseline, and the change along each transect was calculated between each successive shoreline.**"

As a result of this change, we have removed L188-195 and Figure A2.

The authors state that they manually identified the boundaries for the lower and upper shorelines from ICESat2 data. Can you provide some information in the main text about the criteria used to delineate these?

We have updated the text to clarify how these boundaries were delineated

" We manually identified the point corresponding to the backshore/onshore boundary (henceforth referred to as the "upper shoreline") and backshore/beach boundary (the "lower shoreline") **based on the visual breaks in the along-track slope**"

To what extent could you use other geospatial data products to help interpret these results? This may be especially helpful if you are concerned that ShoreZone is out of date given the rates of erosion here. The Jorgenson 2014 maps of thermokarst, ice content, etc. may be especially helpful and can be directly overlain on the modern landscape/shoreline, or Lara et al., 2018 landform mapping. References below.

We thank the reviewer for bringing these data products to our attention. We note that the Jorgenson et al. (2014) database classifies our entire study region as one unit, such that it doesn't provide any additional information on the spatial variability of shoreline characteristics. We have added it as an additional reference when discussing the general setting of our study site.

The Lara et al. (2018) landform provides a higher-resolution and more detailed landform map compared to ShoreZone. While there are a couple coastline-specific classes (such as sand dunes), we feel that an analysis of the specific classes presented here (such as the various types of polygonal terrain) is beyond the scope of this study.

Several typos throughout (some examples):

L103 typo 'strudy' should be 'study'

We have corrected this typo

L156 typo 'only occur when retreat when ocean temperatures'

We've removed "when retreat", which was a typo

L252: spatially averaged shoreline **retreat rate**? Missing 'retreat rate'

We have updated this sentence to: " **The** spatially averaged shoreline **change rate** in our study area was ..."

The Gibbs & Richmond dataset citation is, I believe, incorrect. And the DOI link 'cannot be found.' The 2017 reference should be more appropriate: https://pubs.usgs.gov/publication/ofr20171107

We thank the reviewer for bringing the incorrect doi and updated report to our attention. We have updated the doi link to: https://doi.org/10.3133/ofr20151048. We note that the referenced 2017 report contains updated shoreline statistics for each region, but not in the level of spatial detail needed for the shoreline estimates reported in L107, L335, L347-348, and L355. It also doesn't include the general site descriptions that we reference in section 2.1. Therefore, we feel that it is more appropriate to keep the original 2015 citation.

The referencing is mostly quite thorough, but a few others that I would suggest and have referenced above in individual comments:

Baranskaya A, Novikova A, Shabanova N, Belova N, Maznev S, Ogorodov S and Jones B M 2021 The role of thermal denudation in erosion of ice-rich permafrost coasts in an

Erikson L H, Gibbs A E, Richmond B M, Storlazzi C D, Jones B M and Ohman K A 2020 Changing storm conditions in response to projected 21st century climate change and the potential impact on an arctic barrier island–lagoon system—a pilot study for Arey Island and Lagoon, eastern Arctic Alaska U.S. Geological Survey Open-File Report 2020–1142 p 68

Jorgenson T, Shur Y, Kanevskiy M and Grunblatt J 2014 Permafrost database development—Characterization and mapping for Northern Alaska

Lara M J, Nitze I, Grosse G and McGuire A D 2018 Tundra landform and vegetation productivity trend maps for the Arctic coastal plain of northern Alaska Sci. Data 5 180058

Piliouras A, Jones B, Clevenger T, Gibbs A and Rowland J C 2023 Variability in terrestrial characteristics and erosion rates on the Alaskan Beaufort Sea coast. Env. Res. Letters.

Wobus C, Anderson R, Overeem I, Matell N, Clow G and Urban F 2011 Thermal erosion of a permafrost coastline: improving process-based models using time-lapse photography Arct. Antarct. Alp. Res. 43 474–84

We thank the reviewer for these recommendations, and have updated the text to include these additional citations when referencing drivers and mechanisms of shoreline change in sections 1 and 2. As mentioned above, we feel the Lara et al. paper is outside of the scope of our work.

**Updated Data Analysis**

In response to reviewer comments, we made the decision to re-calculate our shoreline change estimates from both Planet and ICESat-2 in the cross-shore direction instead of the north-south direction. We added the following descriptions of our new methods:

**Section 2.3:** "we estimated shoreline change **using an approach similar to the USGS Digital Shoreline Analysis Software (Himmelstoss et al., 2021). In this workflow, a reference shoreline (hereafter referred to as the baseline) is selected. Cross-shore transects are generated at regular intervals along-shore that are perpendicular to this baseline. The change between consecutive shorelines is then calculated along these transects. We created a baseline by smoothing the raw 2020 shoreline with a 60 m along-shore running mean in order to assure adjacent cross-shore transects did not intersect on the time interval of our study site. Transects were generated every 10 m along the baseline, and the change along each transect was calculated between each successive shoreline.**"

**Section 2.4:** "We identified the intersection between each ICESat-2 track and the corresponding imagery- derived shoreline and compared the shoreline positions and  **cross-shore** retreat estimates derived from Planet and the two ICESat-2 boundaries. **Under the assumption that the majority of the observed position change in our ICESat-2 boundaries is due to cross-shore change, we projected our ICESat-2-derived change estimate into the local cross-shore direction. The local cross-shore vector was based off of the baseline transect (as defined in Section 3.2) that was located closest to the geographic midpoint of the four ICESat-2 transects associated with each ground track.**

In order to characterize the morphology of each profile, we calculated both the backshore elevation and the backshore slope. The backshore elevation was defined as the elevation difference between the backshore/onshore boundary and the mean off-shore (north of the backshore/offshore) elevation from the 02 July 2021 elevation profiles, which we used as a proxy for local sea level (illustrated in Fig. A2). The backshore slope was estimated using a linear fit of all points between the backshore/on-shore and backshore/beach boundaries, **projected into the local cross-shore direction.**"

We removed **L188-195**, **L258-262,** and Figure A2, as they were no longer needed.

These changes resulted in updates to the estimates reported in Table 3, Table 4, and Table A2; as well as Figure 3, Figure 4, Figure 5, and Figure 6. The updated data tables are shown below for reference. We identified a couple of errors in the original values in Table A2, where estimates were based on an intermediate dataset and not the published dataset. Specifically:
- The lower boundary change at Ground Track 2r was listed as -30.1 m when it should have been -20.4 based on our published data. With our updated processing it is now -20.3 m.

- The lower boundary change at Ground Track 2r was listed as 17.5 m when it should have been 15.6 m based on our published data. With our updated processing it is still 15.6 m.

**Table 3.** Estimated shoreline change between each successive image observation date in each of the three regions identified in Figure 3, as well as across the whole study area. The mean and range are listed.

| Year | Region 1 | Region 2 | Region 3 |
|---|---|---|---|
| 2019 | -18.5 m (-29.4m – -7.0 m) | -38.4 m (-58.2 m – -15.7 m) | -0.6 m (-66.7 m – 18.6 |
| 2020 | -13.0 m (-21.6 m – -3.4 m) | -16.8 m (-27.6 m – -3.0 m) | -13.5 (-41.0 m – 6.0 m |
| 2021 | -7.1 m (-14.2 m – 1.6 m) | -17.7 m (-30.9 m – -5.6 m) | -3.9 (-20.7 m – 5.6 m) |
| 2019–2021 | -12.9 m (-29.4 m – 1.6 m) | -24.3 m (-58.2 m – -3.0 m) | -6.0 m (-66.7 m – 18.6 |

**Table 4.** Onshore heights and slopes derived from the identified upper and lower boundaries for each ICESat-2 observation.

| Track | Date | Backshore elevation (m) | Backshore slope (%) |
|---|---|---|---|
| Ground Track 3r | 07 April 2019 | 5.02 | 15 |
| (Region 1) | 04 January 2020 | 4.87 | 28 |
| | 02 July 2021 | 4.89 | 28 |
| | 31 December 2021 | 4.96 | 11 |
| Ground Track 2r | 07 April 2019 | 1.33 | 2.4 |
| (Region 2) | 04 January 2020 | 0.93 | 3.9 |
| | 02 July 2021 | 0.70 | 6.7 |
| | 31 December 2021 | 2.04 | 3.5 |
| Ground Track 1r | 07 April 2019 | 1.89 | 3.6 |
| (Region 3) | 04 January 2020 | 1.67 | 5.9 |
| | 02 July 2021 | 1.64 | 7.0 |
| | 31 December 2021 | 1.64 | 4.3 |

Table A2. North-South change in the upper and lower boundaries estimated from ICESat-2 and the land-water boundaries estimated from Planet at each sampled ICESat-2 location

| Track | Boundary | 2019 | 2020 | 2021 |
|---|---|---|---|---|
| Ground Track 3r | upper boundary (Region 1) |  -16.9 m (± 4.9 m) |  -20.3m (± 5.9 m) |  -6.6 m (± 4.9 m) |
| | lower boundary |  -28.4 m (± 4.9 m) |  -18.4 m (± 5.9 m) |  20.3 m (± 4.9 m) |
| | land-water boundary (Planet) |  -13.7 m (± 3.1 m) |  -12.2 m (± 3.1 m) |  -3.6 m (± 3.1 m) |
| Ground track 2r (Region 2) | upper boundary |  -45.3 m (± 4.3 m) |  -2.5 m (± 4.0 m) |  -62.1 m (± 4.3 m) |
| | lower boundary |  -67.2 m (± 4.3 m) |  -12.5 m (± 4.0) |  -20.3 (± 4.3 m) |
| | land-water boundary (Planet) |  -36.9 m (± 3.1 m) |  -10.8 m (± 3.1 m) |  -27.0 m (± 3.1 m) |
| Ground track 1r (Region 3) | upper boundary |  10.2 m (± 5.6 m) |  1.9 m (± 4.0 m) | 1.7 m (± 5.6 m) |
| | lower boundary |  -7.7 m (± 5.6 m) |  -10.0 m (± 4.0 m) |  15.6 m (± 5.6 m) |
| | land-water boundary (Planet) |  13.8 m (± 3.1 m) |  -1.6 m (± 3.1 m) | 3.8 m (± 3.1 m) |

We note that our updated estimates don't impact our findings.

**Additional Changes**

**L175-176** We made the following additional adjustment to our methods:

"we smoothed each shoreline using a 30 m along-shore running mean and sampled every 10 m  **alongshore**"

**L226-231**: We updated the definition of our morphologic parameters and moved this section to it's own paragraph:

"...We identified the intersection between each ICESat-2 track and the corresponding imagery-derived shoreline and compared the shoreline positions and north-south retreat estimates derived from Planet and the two
ICESat-2 boundaries

In order to characterize the morphology of each profile, we calculated both the backshore elevation and the backshore slope. The
backshore elevation was defined as the elevation difference between the backshore/onshore boundary and the mean offshore
(north of the backshore/offshore) elevation from the 02 July 2021 elevation profiles, which we used as a proxy for local sea
level (illustrated in Fig. A3). The backshore slope was estimated using a linear fit of all points between the backshore/onshore and backshore/beach boundaries, **projected into to the local cross-shore direction**
"

**L344-345** We have rephrased our comparison of our mean retreat rates to Jone et al (2018) on:

"Our mean retreat rates in 2020 and 2021 fall within the range of year-to-year rates observed by Jones
et al. (2018) (-6.7 m/a to -22.6 m/a), whereas our mean retreat in 2019 () **23.7 m**  **slightly** exceeds that range"

We note that -29.0 m was a typo - based on the original data, it should have been -24.0 m.

**Figure 3**: We have updated the caption for 3 b):

" **Cross-shore** shoreline change calculated between successive years."

**Figure 4**: We have updated the caption:

"Positions are given  **as the cross-shore distance from** the Planet-derived shoreline from 25 June 2019"

**Figure 5**: We added trendlines to the legend, and corrected a legend label error in 5a) where gt3r was mislabeled as gt1r

**Figure 6**: the x-axis for subfigures, b), d), and f) have been changed from 'North-South Distance' to 'Cross-Shore Distance'. We have also updated the text in the caption:

"The estimated location of the upper and lower shoreline boundary for each date is marked, along with the  **cross-shore** locations of the Planet-derived shorelines."

 **Figure 7**: The x-axis has been changed to 'Cross-Shore Distance'

**L432-436**: We added some additional text for clarity:

"We see moderate advance (+10.2 m of shoreline change) and a drop in elevation (-0.22 m) of the backshore boundary in 2019, but very little change in either the position (+1.7 m to +1.9 m, which falls within our estimated uncertainty) or elevation (which ranges from 1.64 m to 1.67 m) **of the backshore** over the next 2 years. Fluctuations in the slope between observation dates are driven primarily by changes in the lower boundary (**which undergoes between -10.0 and +15.6 m of shoreline change per year)**, which is consistent with sediment deposition and removal at the beach in front of the lagoon or changes in snow cover."

**Table A2**: A formatting error where the "(Region 1)" label was placed in the "" was been corrected (see new table below):

**Table A2.** North-South change in the upper and lower boundaries estimated from ICESat-2 and the land-water boundaries estimated from Planet at each sampled ICESat-2 location

| Track | Boundary | 2019 | 2020 | 2021 |
|---|---|---|---|---|
| Ground Track 3r (Region 1) | upper boundary | -16.9 m (± 4.9 m) | -20.3m (± 5.9 m) | -6.6 m (± 4.9 m) |
| | lower boundary | -28.4 m (± 4.9 m) | -18.4 m (± 5.9 m) | 20.3 m (± 4.9 m) |
| | land-water boundary (Planet) | -13.7 m (± 3.1 m) | -12.2 m (± 3.1 m) | -3.6 m (± 3.1 m) |
| Ground track 2r (Region 2) | upper boundary | -45.3 m (± 4.3 m) | -2.5 m (± 4.0 m) | -62.1 m (± 4.3 m) |
| | lower boundary | -67.2 m (± 4.3 m) | -12.5 m (± 4.0) | -20.3 (± 4.3 m) |
| | land-water boundary (Planet) | -36.9 m (± 3.1 m) | -10.8 m (± 3.1 m) | -27.0 m (± 3.1 m) |
| Ground track 1r (Region 3) | upper boundary | 10.2 m (± 5.6 m) | 1.9 m (± 4.0 m) | 1.7 m (± 5.6 m) |
| | lower boundary | -7.7 m (± 5.6 m) | -10.0 m (± 4.0 m) | 15.6 m (± 5.6 m) |
| | land-water boundary (Planet) | 13.8 m (± 3.1 m) | -1.6 m (± 3.1 m) | 3.8 m (± 3.1 m) |

**Figures A3, A4, and A5**: The x-axis has been changed to "Cross-Shore Distance"

---

## Referee Report (RR1)

**Review of**
**Multiple modes of shoreline change along the Alaskan Beaufort Sea**
**observed using ICESat-2 altimetry and satellite imagery**
**Bryant et al. 2024-1656**
**#2**

I am satisfied with the answer that the authors provided to my comments.

I particularly appreciated that they added the suggested Figure A7.

I am only left with small concerns which should not prevent publication of the article, nor another round of review.

     1. Code should be shared.

     2. It is quite intriguing that the Planet-derived shoreline falls between the ICESat-2 upper and lower shoreline. It really makes me curious to see a quick comparison with external data (e.g. Sentinel, Landsat) to, maybe, rule out systematic geolocation error in, at least, one of the products.

     3. I do not understand why the increase of coastal reatreat since 2000 is not believed to be an actual signal. It is true that measurements periods are shorter, but they are consitstent with Jones et al. (2018) and averaged together, they show an acceleration compared to similar period duration of Jones et al. (2009). I would repeat that clearly in conclusion as it is an interesting finding of this work.

L17: "complimentary"?

L60 "10m" missing a space

L96: "$m\ a^{-1}$" The author's guide does not seem to require units to be in italic

L113 "and evidence that retreat rates are increasing in recent years"

L122 "CNES **A**irbus"

L143 "thermal mechanical abrasion" thermo-mecanical? Thermal and mechanical?

L213 "3r,2r" missing space

L232 "the background-corrected standard deviation" thanks for adding more details about this error. However, this term remains unclear. "Background" is never used elsewhere in the text. And "standard deviation" of what?

L286 "a position change estimate uncertainty" thanks for adding "position", maybe "estimate" can now be deleted?

L297 "in 2020 (-16.8 m) **and** in 2021 (-17.7 m)"

L302 "**We find that** 24%…" to avoid starting a sentence with a number.

L408 "characterized" twice in the same sentence

L431 "that that"

L437 "(-20.3 m to - 67.2 m) change" move the numbers after "change"? Or even after "boundaries"

Figure 6 "SlideRulederived" missing "-" or a space.

"overlain"

Figure 7 This figure could be about twice smaller. Text size is huge at the moment.

Figure A7 is not cited in the text, it could fit in 4.1.

---

## Author Response (AR2)

Review comments are in black
Authors' responses are in blue, with proposed **new text** in bold. All line numbers refer to line numbers in the original/unrevised manuscript.

**Referee #1: César Deschamps-Berger**
I am satisfied with the answer that the authors provided to my comments.
I particularly appreciated that they added the suggested Figure A7.
I am only left with small concerns which should not prevent publication of the article, nor another round of review.

1. Code should be shared.
   We have created a repository and archived it on Zenodo:
   https://doi.org/10.5281/zenodo.14777816

2. It is quite intriguing that the Planet-derived shoreline falls between the ICESat-2 upper and lower shoreline. It really makes me curious to see a quick comparison with external data (e.g. Sentinel, Landsat) to, maybe, rule out systematic geolocation error in, at least, one of the products.
   While we agree that a comparison with additional datasets would be informative, we feel that it would be best to not introduce a new dataset at this stage. We also note that given the published uncertainties of each dataset are not large enough to fully explain the discrepancies. We have updated the text as follows:

   "**Given the published geolocation errors from ICESat-2, (up to 4.8 m, Luthcke et al. (2021)) and Planet (< 10 m, Planet Team (2023)), the difference between ICESat-2 and Planet derived shorelines is not thought to be due to a consistent geolocation offset between the two satellite estimates**. The difference could be explained by a consistent landward bias in our NDWI thresholding technique (Section 2.3) or differences introduced by snow cover and variations in the local water level."

3. I do not understand why the increase of coastal reatreat since 2000 is not believed to be an actual signal. It is true that measurements periods are shorter, but they are consitstent with Jones et al. (2018) and averaged together, they show an acceleration compared to similar period duration of Jones et al. (2009). I would repeat that clearly in conclusion as it is an interesting finding of this work.
   We have rephrased this paragraph to better distinguish between the comparison of our rates to short-term recent rates (2002 onwards) and long term historical (pre 2002) rates:

   "Our estimates of spatially averaged (regional) mean annual shoreline change (-10.5 m a−1 to -23.7 m a−1) across our study region are higher than long-term historical estimates and similar to recent observations (Fig. A7). Gibbs and Richmond (2015) estimated a regional mean of -6.3 m a−1 and a local maximum of -18.6 m a−1 between Drew Point and Cape Halket between 1947 and 2002. Jones et al. (2009) estimated shoreline change across this region over multiple

time intervals and found -6.8 m a−1 of change between 1955 and 1979, -8.7 m a−1 between 1979 and 2002, and -13.6 m a−1 between 2002 and 2007. A follow-up study by Jones et al. (2018) estimating shoreline change over a 9 km region covering our study area found a 10-year mean shoreline change rate of -17.2 m a−1 between 2007 and 2016. In addition to estimating a 10-year mean, Jones et al. (2018) reported regional year-to-year rates, which ranged from -6.7 m a−1 to -22.6 m a−1. Our mean retreat rates in 2020 and 2021 fall within the range of year-to-year rates observed by Jones et al. (2018) whereas our mean retreat in 2019 (-23.7 m) slightly exceeds that range. Our 3-year mean of observed retreat rates (-16.5 m) is similar to the decadal-scale estimate from Jones et al. (2018), and consistent with an increase in local shoreline change rates compared to 2002-2007 (Jones et al., 2009). Our retreat estimates and the post-2002 estimates of Jones et al. (2009) and Jones et al. (2018) are all higher than the pre-2002 decadal-scale estimates of Gibbs and Richmond (2015) and Jones et al. (2009). This increase could be reflective of a long-term increase in retreat rates, although it may also be due to differences in time scales between studies, as short-term estimates of shoreline change tend to be higher in magnitude than long-term estimates (Sadler and Jerolmack, 2015)."

L17: "complimentary"?
We have corrected this to "complementary"

L60 "10m" missing a space
We have added a space

L96: "m a−1" The author's guide does not seem to require units to be in italic
We thank the reviewer for bringing this to attention, and have removed the italics for all units in the text and figure A7.

L113 "and evidence that retreat rates are increasing in recent years"
L122 "CNES Airbus"
We have capitalized "Airbus"

L143 "thermal mechanical abrasion" thermo-mecanical? Thermal and mechanical?
We have updated this to "thermal **and** mechanical"

L213 "3r,2r" missing space
We have added a space

L232 "the background-corrected standard deviation" thanks for adding more details about this error. However, this term remains unclear. "Background" is never used elsewhere in the text. And"standard deviation" of what?

We have updated the text to clarify the terms used and refer readers to Smith et. al for the full definition:

"The photon elevation error is assumed to be uniform for a given segment, and is estimated as the maximum of the segment RMS error and the  **robust spread of photons as defined by Smith et al. (2019)**. **This robust spread is based on the vertical distribution of signal photon heights and the estimated rate of background photons (i.e., photons that are not surface signals).**"

L286 "a position change estimate uncertainty" thanks for adding "position", maybe "estimate" can now be deleted?
We agree and have removed "estimate"

L297 "in 2020 (-16.8 m) and in 2021 (-17.7 m)"
We have added "and" per this suggestion.

L302 "We find that 24%…" to avoid starting a sentence with a number.
We have added "**We found that**…" to the beginning of the sentence

L408 "characterized" twice in the same sentence
To avoid this redundancy, we updated this sentence to:

"The eastern edge of Region 3 is characterized as inundated tundra, which refers to areas  **where** thaw subsidence and surface ponding **are present**"

L431 "that that"
We have removed the extra "that"

L437 "(-20.3 m to - 67.2 m) change" move the numbers after "change"? Or even after "boundaries"
We have moved the number after "change":

"The changes in slope between 2019 and 2020 and between early and late 2021 were driven by high and variable  change **(-20.3 m to - 67.2 m)** in both the upper and lower boundaries…"

Figure 6 "SlideRulederived" missing "-" or a space.
"Overlain"
We have updated this sentence in the caption to:

"ATL03 photon clouds from each ICESat-2 pass, with the SlideRule-derived (ATL06-SR) elevation profile overlai**d on the photon clouds** for…"

Figure 7 This figure could be about twice smaller. Text size is huge at the moment.
We have reduced the figure width from 0.9 to 0.6 of the columnwidth. We have also slightly reduced the font size of the title and axes.

Figure A7 is not cited in the text, it could fit in 4.1.
We thank the reviewer for bringing this to our attention, and have added it to the first sentence of 4.1 (L361):

"Our estimates of spatially averaged (regional) mean annual shoreline change (-10.5 m a$^{-1}$ to -23.7 m a$^{-1}$) across our studyregion are higher than long-term historical estimates and similar to recent observations **(Fig A7)**."

**Other edits**

-We have added copyright statements to the captions of Figure 1, Figure 2, Figure 3 and Figure 6.

-L195: added additional explanation and a reference for our transect generation process:

"Transects were generated every 10 m along the baseline **using a modified version of the SDS_transects routine (Vos, 2024) from the CoastSat Toolbox (Vos et al., 2019) that generates transects that bisect the shoreline**The change along each transect was calculated between each successive shoreline."

Vos, K.: SDS_transects.py, https://github.com/kvos/CoastSat/blob/master/coastsat/SDS_transects.py, 2024.

-L206-207: Added the following clarification:
"We calculated the  **cross-shore difference** between each shoreline position and the mean position of its cluster and pooled the residuals across all shorelines"

-L46, L95, L403, L404: replaced "coastline" with "shoreline" for consistency

-L317-320: updated sentence for clarity:
"However, we found that the Planet-derived land-water boundary consistently falls landwards **(south)** of the ICESat-2-derived lower boundary (by 8.6 ± 4.2 m to 41.1 ± 4.5 m) and either seawards (by up to to 36.0 m ± 4.5 m) or slightly landward of the ICESat-2-derived upper boundary (by up to 3.3 ± 4.0 m) (Fig. 4), **such that it is located on the backshore or onshore section of the elevation profiles.**"

-Corrected the following typos:
L448:  35.3
L450 :  8th

-L514: rephrased sentence:
"Overall, we found that annual retreat rates from both datasets are  **consistent with recent** estimates of shoreline change over the last decade"

-Replaced the data availability statement with a code availability statement

-Updated Acknowledgements

-Table A1: The standard deviation of individual clusters were updated based on the transition from north-south to cross-shore distance. We note that all changes were <= 0.1 m, and that the final uncertainty estimate did not change within the reported precision.

-Updated the following references:

-
Arndt, P. S. and Fricker, H. A.: A framework for automated supraglacial lake detection and depth retrieval in ICESat-2 photon data across the Greenland and Antarctic ice sheets, The Cryosphere, 18, 5173–5206, https://doi.org/10.5194/tc-18-5173-2024, 2024.

Smith,  **Adusumilli, S., Csathó, B. M., Felikson, D., Fricker, H. A., Gardner, A., Holschuh, N., Lee, J., Nilsson, J., Paolo, F. S., Siegfried, M. R., Sutterley, T** and the ICESat-2 Science Team.: ATLAS/ICESat-2 L3A Land Ice Height, Version 6, https://doi.org/10.5067/ATLAS/ATL06.006, 2023.

Timmermans, M. L. and Labe, Z.: NOAA Arctic Report Card 2023: Sea Surface Temperature, https://doi.org/10.25923/E8JC-F342, publisher: , 2023.